# The Formation of Supramolecular Chiral Materials from Achiral Molecules Using a Liquid-Crystallin System: Symmetry Breaking, Amplification, and Transfer

Atsushi Yoshizawa

Department of Frontier Materials Chemistry, Graduate School of Science and Technology, Hirosaki University, 3 Bunkyo-cho, Hirosaki 036-8561, Aomori, Japan; ayoshiza@hirosaki-u.ac.jp

**Abstract:** Recently, the formation of chiral materials by the self-organization of achiral small molecules has attracted much attention. How can we obtain chirality without a chiral source? Interesting approaches, such as mechanical rotation, circularly polarized light, and asymmetric reaction fields, have been used. We describe recent research developments in supramolecular chirality in liquid crystals, focusing primarily on our group's experimental results. We present the following concepts in this review. Spontaneous mirror symmetry breaking in self-assembled achiral trimers induces supramolecular chirality in the soft crystalline phase. Two kinds of domains with opposite handedness exist in non-equal populations. The dominant domain is amplified to produce a homochiral state. Chirality is transferred to a polymer film during the polymerization of achiral monomers by using the homochiral state as a template. Finally, we discuss how the concepts obtained from this liquid crystal research relate to the origin of homochirality in life.

**Keywords:** chirality; symmetry breaking; chiral transfer; liquid crystals; helical polymers





## 1. Introduction

Generally, producing chiral materials requires a chiral source, such as chiral building blocks or chiral catalysts. Recently, the formation of chiral architectures by the self-organization of achiral small molecules has been intensively investigated [1–4]. How does chirality occur? How is chirality transferred to a higher-ordered architecture? Investigating these mechanisms is significant not only for producing chiral materials without a chiral component but also for elucidating the origin of homochirality in life [5,6]. Spontaneous mirror symmetry breaking produces chirality in an achiral system. Symmetry breaking via consecutive amplification has been intensively investigated [7]. Among them, self-amplification of symmetry breaking by catalyst is an attractive approach [8–11]. Soai et al. discovered the amplification of enantiomer excess (ee) during the asymmetric autocatalysis of 5-pyrimidyl alkanol after the addition of $i$-Pr$_2$Zn to pyrimidine-5-carbaldehyde [12,13]. Even pyrimidyl alkanol with as low as ca. $5 \times 10^{-5}$% ee enhances its ee to >99.5% ee during asymmetric autocatalysis. Furthermore, Soai et al. demonstrated that enantiomerically enriched pyrimidyl alkanol was obtained stochastically from achiral 2-alkynyl-pyrimidine-5-carbaldehyde and $i$-Pr$_2$Zn without adding chiral compounds [13,14]. The racemate does not contain exactly the same numbers of each enantiomer based on the theory of statistics. Therefore, when a system involves asymmetric autocatalysis with an amplification of enantiomer excess, an initial bias in ee resulting from a fluctuation in the racemate would be enhanced to produce an enantiomeric product by the asymmetric autocatalysis, as shown in Figure 1 [13]. This synthetic procedure produces enantioenriched compounds without adding chiral substances. It is spontaneous absolute asymmetric synthesis. This is the induction of chirality at a molecular level.

**Figure 1.** Asymmetric synthesis of pyrimidyl alkanol without adding chiral substances. Reprinted with permission from ref. [14]. Copyright 2003 Elsevier.

Spontaneous mirror symmetry breaking has been known to occur under mechanical rotations. Kondepudi et al. found that the stirred crystallization of $NaClO_3$ resulted in spontaneous mirror symmetry breaking [15]. The achiral compound $NaClO_3$ crystallizes in enantiomeric forms. Similar enantiomeric crystallization was observed in stirred 1,1′-binaphthyl melt [16,17]. The mechanical-rotation-induced chirality of supramolecules has been investigated. Recently, Ishi et al. reported a rotary-evaporation-induced enantioselective aggregation of achiral phthalocyanines [18] (Figure 2). The chiral films were obtained on the bottom of the flask. Interestingly, the handedness of the chirality depends on the rotational direction of the evaporator. This is the induction of chirality at a supramolecular level.

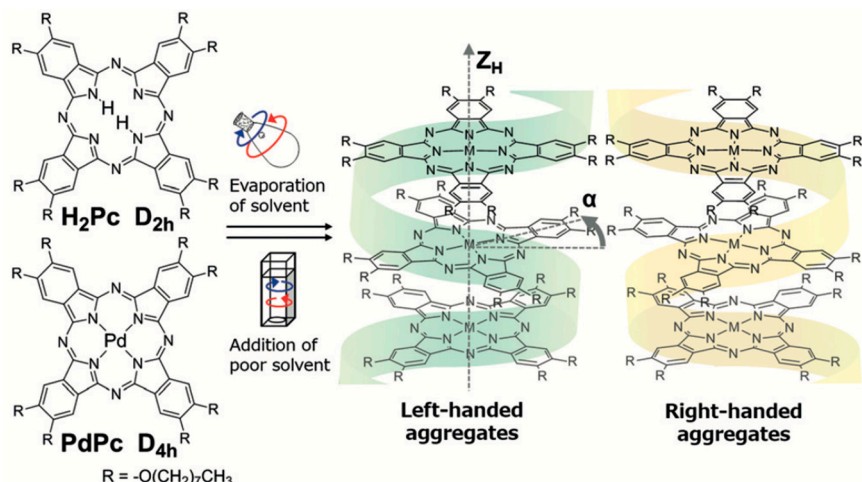

**Figure 2.** Molecular structures of phthalocyanines ($H_2Pc$ and $PdPc$) and chiral H-aggregates of phthalocyanines prepared by mechanical rotations, which were either rotary evaporation or magnetic stirring. Blue and red arrows show clockwise and anticlockwise rotations, respectively. Reprinted with permission from ref. [18]. Copyright 2019 WILEY-VCH Verlag GmbH & Co. KGaA, Weinheim, Germany.

External fields, such as photonic and electronic fields, play an important role in symmetry breaking. Circularly polarized light (CPL) can induce homochirality in an achiral molecular assembly [19]. CPL is a chiral electromagnetic radiation. The handedness of the induced chirality is attributed to that of CPL. Orio et al. reported the chiral induction in an achiral nematic glassy phase of azobenzene polymer **P100** by irradiating CPL [20]. The photoinduced supramolecular chirality could begin at the molecular level with asymmetric photoisomerization due to the chiral exciting right. The asymmetric chiral modification induced by CPL leads to the conversion of a glassy nematic phase into a glassy chiral

nematic phase. The glassy nematic and glassy chiral nematic phases are glassy phases with a nematic-like orientational order and a chiral nematic-like orientational order, respectively. Both phases have a positional order. The CPL irradiation induces the asymmetric photoisomerization of an azobenzene group in the side chain. The authors noted that irradiation creates a significant population of the Z-isomer, and this can play a role as a plasticizer. The helical handedness can be switched using orthogonal CPL. The transfer of chirality from CPL to azopolymer through chiral conformation is proposed as a model for explaining the supramolecular chirality (Figure 3) [20].

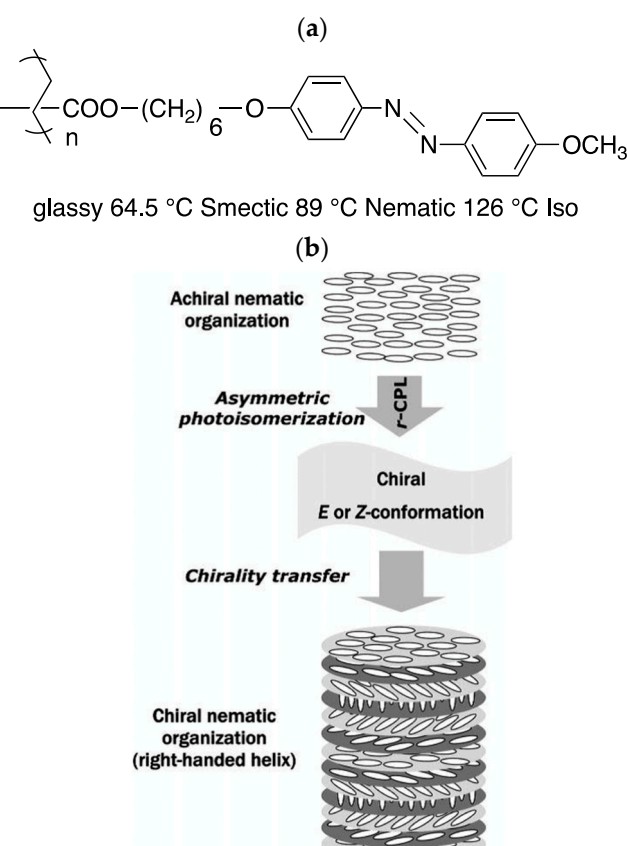

**Figure 3.** (**a**) Structure and thermal properties of the side-chain liquid crystal azopolymer **P100**. (**b**) Proposed model for the photoinduction of chirality in azo materials. Reprinted with permission from ref. [20]. Copyright 2007 WILEY-VCH Verlag GmbH & Co. KGaA, Weinheim, Germany.

Spontaneous mirror symmetry breaking has been observed not only in liquid-crystalline and soft-crystalline phases but also in isotropic liquids [21–38]. Most of them exhibit chiral conglomerates consisting of domains with opposite handedness. For example, helical nanofilament phases (HNF phases, also assigned as B4 phases) are described here. They have a characteristic helical network, and they act as a porous nanoconfinement medium of large internal area, with the guest material confined to nanoscale interstitial volumes between the filaments [31]. Figure 4a shows a typical molecular structure of the bent-core molecule exhibiting a soft-crystalline helical nanofilament phase. Figure 4b shows the optical textures in the HNF phase under crossed and uncrossed polarizers. The texture under crossed polarizers was nearly dark. By observing the sample under slightly uncrossed polarizers, the texture was split into darker and brighter domains. By uncrossing the polarizers in opposite directions by the same angle, the darker and brighter domains were exchanged. These results indicate that they have optical activity with opposite handedness. Figure 4c shows a freeze-fracture transmission electron microscopy (FF-TEM) image of the HNF phase and a schematic model of the helical network. Such bent-core molecules themselves do not have a chirality at a molecular level. There are two proposed

mechanisms for the origin of the optical activity. The first is layer chirality [39]. The second is the conformational twist [23]. These mechanisms will be explained later. However, the problem of how we can produce a homochiral phase still remains.

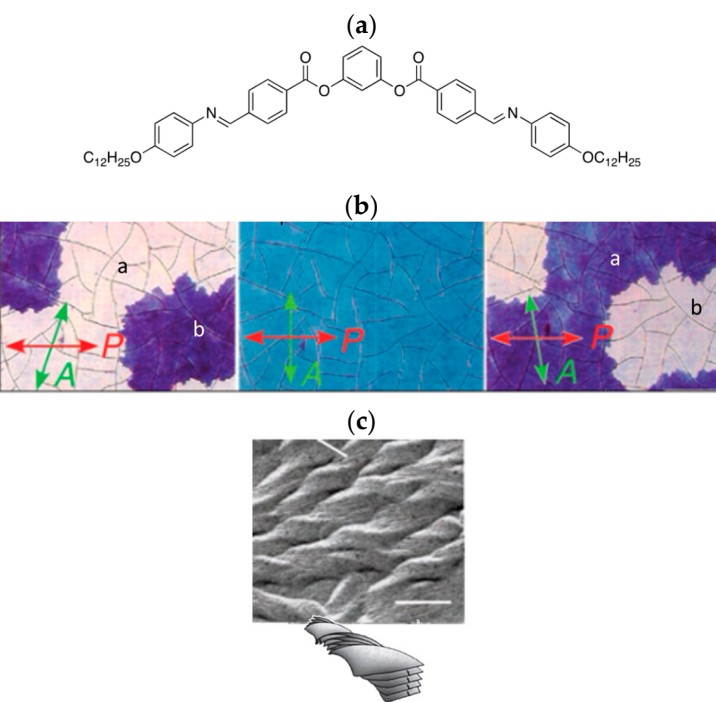

**Figure 4.** (**a**) Molecular structure of a bent-core molecule exhibiting the HNF phase. (**b**) Polarized optical textures in the HNF phase under uncrossed and crossed polarizers. (**c**) FF-TEM image of the HNF phase with its model. Reprinted with permission from ref. [31]. Copyright 2009 American Association for the Advancement of Science America.

Recently, some chiral polymers have been prepared from achiral components or racemates [1,4,40–45]. How does chirality appear in a higher-ordered structure composed of achiral components? Chiral amplification by 'Sergeants and Soldiers effects' or "Majority rules" produces a single helical structure in the polymer from achiral monomers [41]. Helix-sense-sensitive polymerization proposed by Yashima et al. is an interesting and effective method for the synthesis of chiral polymers composed of desired monomer units [2,42]. Furthermore, chiral polymers have been obtained from achiral units under the influence of noncovalent interactions, such as molecular chirality of a catalyst, CPL, and chiral solvents [4]. Helical polyacetylene was synthesized in chiral nematic (N*) liquid crystals [46,47]. The asymmetric reaction field for the acetylene polymerization was prepared by dissolving the Ziegler–Natta catalyst, Ti(O-*n*-Bu)$_4$-AlEt$_3$, into the N*-LC. The N*-LC includes an axially chiral binaphthyl derivative, or asymmetric carbon-containing chiral compound (Figure 5). The molecular chirality of the chiral dopant transforms the achiral nematic phase to the supramolecular chiral nematic phase with the macroscopic helical ordering. The supramolecular chirality transfers to the helical polyacetylene in the course of polymerization of achiral reactive acetylene monomers in the chiral nematic phase. The detailed investigations reveal that left-handed and right-handed H-PA chains are formed in (R)- and (S)-chiral nematic liquid crystals, respectively, and that these helical chains are bundled through van der Waals interactions to form helical fibrils with screw directions opposite to those of the chiral nematic liquid crystals. The bundles of fibrils form the spiral morphology with various sizes of domains. (Figure 6).

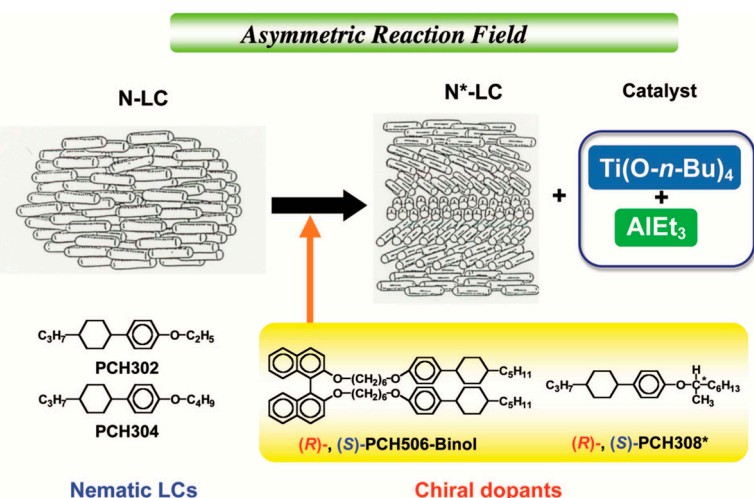

**Figure 5.** Asymmetric reaction field for the formation of helical polyacetylene. Reprinted with permission from ref. [47]. Copyright 2009 American Chemical Society.

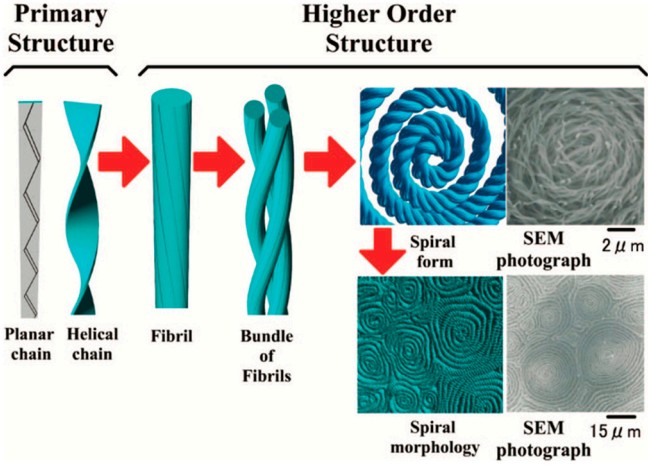

**Figure 6.** Hierarchical helical structures from primary to high order in helical polyacetylene. Reprinted with permission from ref. [47]. Copyright 2009 American Chemical Society.

As described above, asymmetric environments, such as asymmetric rotational directions, CPL, and macroscopic helical ordering, play an important role in the induction of supramolecular chirality except for the symmetry breaking of liquid crystals. In this focused review, after introducing our design concept of liquid-crystalline supermolecules, we describe spontaneous mirror symmetry breaking in achiral liquid-crystalline compounds and the formation of helical polymers using a template with a supramolecular chirality.

## 2. Design Concept of Liquid-Crystalline Supermolecules

On the basis of driving force, liquid crystals are classified as lyotropic and thermotropic liquid crystals. The appearance of lyotropic LC phases of a surfactant-solvent system depends on the solvent concentration. The amphiphilic molecules self-organize into supramolecular assemblies in order to increase their surface areas, which can interact with solvent molecules. On the other hand, thermotropic LCs appear at a certain temperature range. The most common phases are the nematic and smectic phases. A nematic phase has the same flowing character as the isotropic liquids; on the other hand, smectic phases with a layer structure lose such a liquid-like fluidity. Anisotropic intermolecular interactions, i.e., hard-core repulsion and electrostatic interactions, are responsible for the stabilization of the thermotropic liquid-crystalline phases. It is well known that the excluded volume effects determined by hard-core repulsion require a rod-like molecular shape with a length (*L*) to

diameter (*D*) ratio of larger than about 3 for the stabilization of rod-like liquid crystals. In compounds possessing polar groups, the interaction of permanent and induced electrical dipoles contributes to stabilizing LC phases [48]. Chemists design LC molecules with these factors. They usually visualize the static state of the molecule. However, molecules in LC phases have two faces. The static molecular structure can be easily drawn, while the dynamic properties, such as rotation around the long axis, reorientation of the long axis, internal motions, and cooperative motions, are difficult to image. Please see Figure 7.

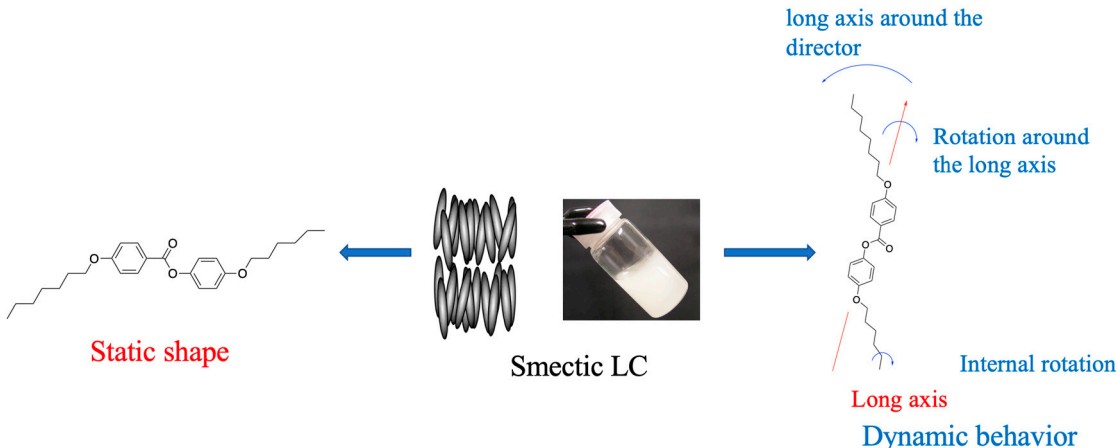

**Figure 7.** Molecules in LC phases have two faces, i.e., static and dynamic images.

We investigated the dynamic behavior of molecules in some smectic phases using solid-state $^{13}$C NMR [49,50]. The NMR studies reveal the following pictures: (1) cooperative motion for the core parts contributes to the orientational order of the molecules in each layer; (2) interlayer permeation of tails causes the correlation between cores adjacent layers; and (3) molecular deformation occurs near the smectic A–chiral smectic C (or smectic C) transition. An illustration of the model is shown in Figure 8. The correlation of molecular motion neighboring molecules considerably affects long-range order in the smectic phases. Goodby noted that the model is also applicable to antiferroelectric and twist grain boundary phases in his paper [51]. Based on the NMR studies, we designed U-shaped and linear-shaped liquid-crystalline oligomers [52]. They are regarded as a supermolecule in which several mesogenic units are connected via flexible spacers. They have short-range order within a single molecule. The short-range order can affect their molecular organization. A U-shaped molecule in which dynamics of the two mesogenic parts correlate mutually can induce a smectic layer ordering in the nematic phase. Some U-shaped compounds stabilized cybotactic N phases (Figure 9) [53]. On the other hand, a linear-shaped dimesogenic molecule has two core parts, which are connected via a flexible spacer. The linear-shaped molecules are expected to enhance the interlayer correlation. We designed a chiral linear-shaped dimesogenic molecule in which two phenylpyrimidine units are connected via optically active 3-methyladipic acid [54]. The molecule itself does not have a large helical twisting power. However, some binary mixtures of the linear-shaped chiral dimer and a host smectic liquid crystal exhibited two kinds of twist grain boundary (TGB) phases [54]. TGB phases were predicted by de Gennes [55] and later by Renn and Lubensky [56] to be a result of the competition between bend or twist deformations and the desire for molecules to form a layer structure. The first experimental observation of a TGB phase was reported by Goodby et al. [57,58]. It was shown to exhibit blocks of SmA molecular order mediated by screw dislocations. It is referred to as the TGBA phase. The appearance of a TGBA phase was explained in terms of twist and bend distortions caused by large molecular fluctuations [57]. TGB phases have been usually observed for rigid compounds with high twisting power. The appearance of the TGB phases in the binary system is explainable in terms of an intercalated chirality [54]. Competition between

twisting force and the desire for molecules to form a layer structure causes a helical ordering of smectic blocks. The chiral dimer might intercalate to adjacent layers composed of achiral hosts. The intercalated dimer can cause a strong correlation of motion and directions between cores in adjacent layers. The molecular chirality of the guest dimer is synchronized via the cooperative motion of the host molecules to produce the TGB phases (Figure 10). In this review, we focus on achiral liquid crystal trimers producing a supramolecular chirality.

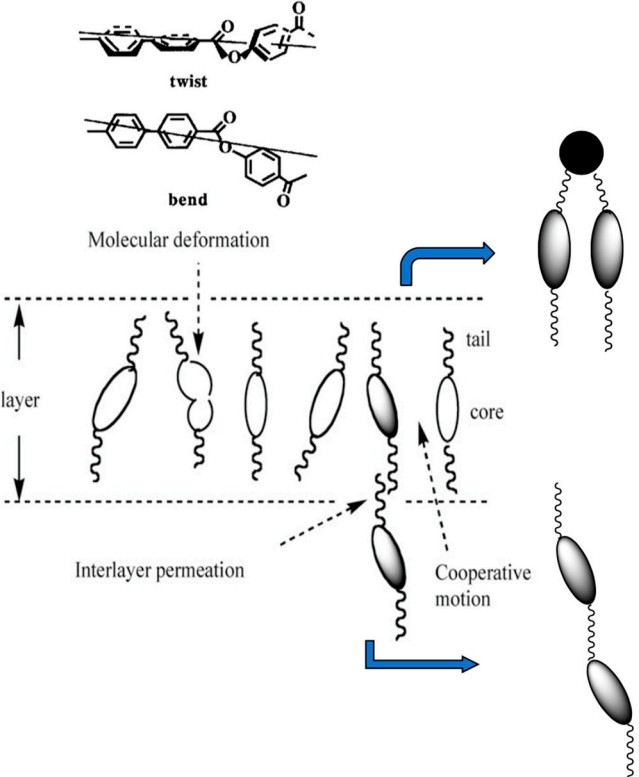

**Figure 8.** Molecular design of supermolecules based on the $^{13}$C NMR studies. The black lines indicate the long axis of each molecule.

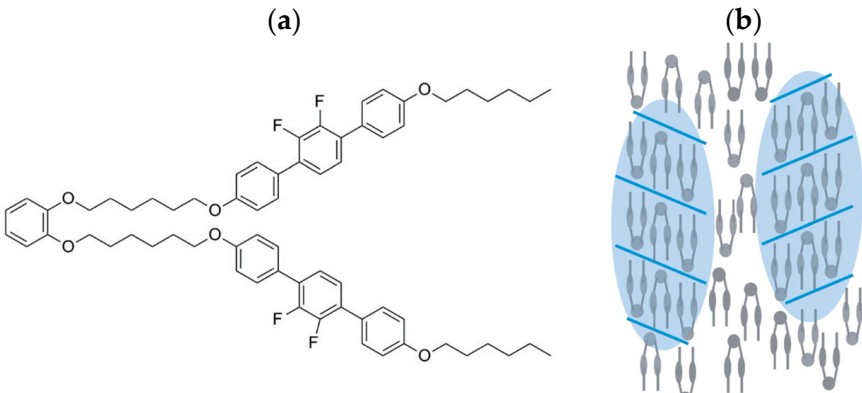

**Figure 9.** (**a**) Molecular structure of the U-shaped compound. (**b**) Schematic model for the nematic phase with the smectic-like layered structure [53].

**Figure 10.** Intercalated chirality model [54].

## 3. Spontaneous Mirror Symmetry Breaking of Achiral Liquid Crystals Possessing a Layer Structure

Since the discovery of a liquid-crystalline phase in cholesterol derivatives, molecular chirality has been necessary to produce a chiral liquid-crystalline phase. Niori et al. discovered ferroelectric switching [21], and then, Sekine et al. suggested chirality in a fluid smectic B2 phase of achiral bent-core molecule **P-n-O-PIMB** [22]. Since these discoveries, a lot of research on polar order and chiral superstructures of liquid crystals has been performed. There are two proposed mechanisms for the origin of optical activity, i.e., layer chirality [39] and conformational chirality [23]. Two bent-core molecules forming a tilted layer structure are shown in Figure 11a. For the molecule on the left, the convex direction and tilt direction of the molecule are clockwise when rotated around the layer's normal direction. On the other hand, the molecules on the right are anticlockwise. Therefore, these two tilted molecules in a layer are chiral. This mechanism is called "layer chirality". With respect to the conformational chirality (Figure 11b), the bent-core molecules have twisted side wings, producing an asymmetric axis.

Spontaneous mirror symmetry breaking observed in optically isotropic layered phases has attracted much attention because of their characteristic three-dimensional structures. They are classified into three categories depending on the local structures, i.e., liquid-crystalline sponge phase [30], helical nanofilament phase [31], and helical nanocrystalline (HNC) phase [32]. Their schematic images are shown in Figure 12. They prefer to form saddle splay curvature locally. Almost all molecules exhibiting the layered chiral conglomerate phases have a rigid bent-core structure [59], except in a few cases in which the HNF phase of an achiral dimer possessing an odd-numbered spacer was reported [60]. Such an inherent rigid bent shape is essential for the formation of these phases. The conformational chirality seems to be more realistic for explaining the formation of twist layer structures than the layer chirality. The inherent molecular twist is synchronized to organize the supramolecular chirality of achiral molecular aggregation. Recently, helical structures were found to arise in some nematic phases [35–37,61–67]. The twist–bend nematic ($N_{TB}$) phase showing domains with opposite handedness was observed for members of achiral $\alpha,\beta$-bis-4-(40-cyanobiphenyl)alkanes with flexible odd-numbered methylene spacers [35]. Figure 13 shows a schematic image of the formation of the $N_{TB}$ phase. Bent-shaped molecules cannot fill the space in the nematic phase. Some molecules rise to form helical structures. The

molecular long axis is tilted with respect to the helical axis in the $N_{TB}$ phase. Figure 14 shows the molecular structures of typical bent-core molecules and a flexible dimer.

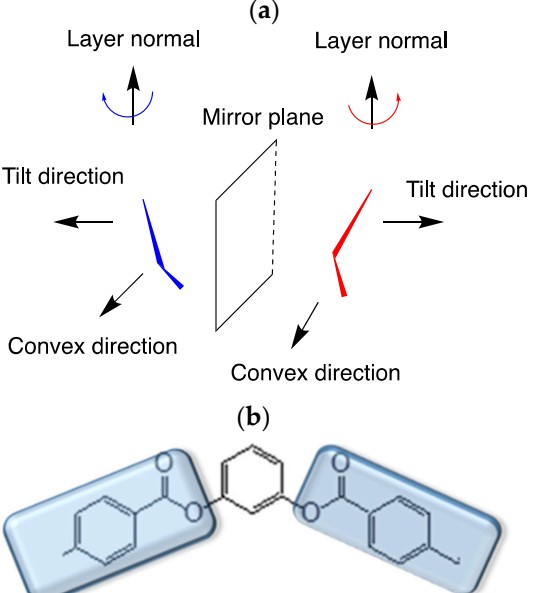

**Figure 11.** (**a**) Layer chirality [39] and (**b**) conformational chirality [23].

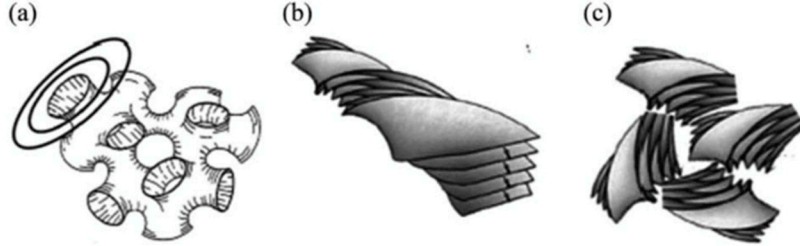

**Figure 12.** (**a**) Sponge phase, (**b**) helical nanofilament phase, and (**c**) helical nanocrystalline phase. The pictures are taken from ref. [59].

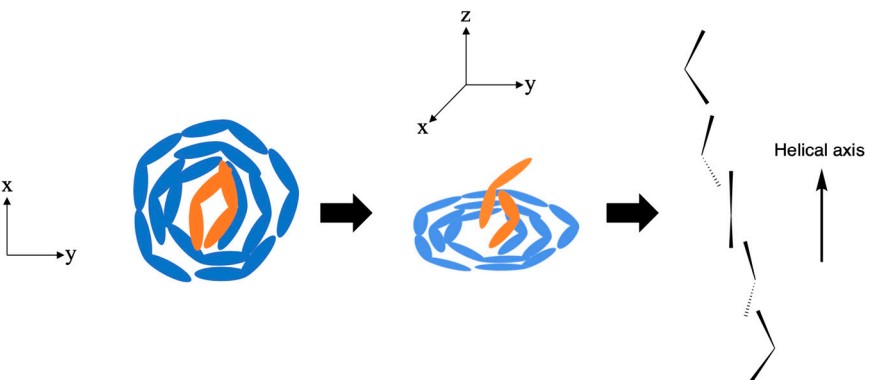

**Figure 13.** Schematic illustration for the formation of the $N_{TB}$ phase.

Sponge phase

HNF phase

HNC phase

$N_{TB}$ phase

**Figure 14.** Bent-shaped LCs exhibiting the layered chiral conglomerate phases and a flexible dimer exhibiting the twist–bend nematic phase. Sponge phase [30]; HNF phase [21,31]; HNC phase [32]; $N_{TB}$ phase [35].

We reported achiral liquid crystal trimers exhibiting chiral conglomerates in the soft-crystalline dark phase possessing a layer structure [68]. We call this dark chiral conglomerate phase DC (dark chiral conglomerate) phase. The phase diagram for a homolog series of trimers **I-(*n,m*)** is shown in Figure 15. The trimers possessing longer and odd-numbered spacers exhibited a phase sequence of Iso–N–DC.

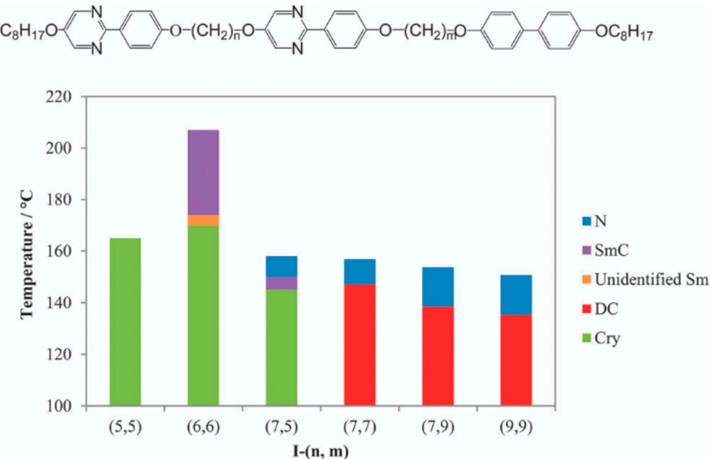

**Figure 15.** Phase diagram on cooling for a homolog series of trimers **I-(*n,m*)**. The picture is taken from ref. [68].

Trimer **I-(7,7)** exhibited a nematic to DC phase transition with a large enthalpy change, as shown in Figure 16. The $\Delta S/R$ value was 10.4. Therefore, the DC phase is not a liquid crystal but a soft crystal. Figure 17 shows the polarized optical textures of the trimer under uncrossed and crossed polarizers. The texture of the DC phase under crossed polarizers

was nearly dark, suggesting that it is optically isotropic. Observing the sample in the DC phase under slightly uncrossed polarizers indicates that the darker and brighter domains have optical activity with opposite handedness. No chiral nature in the N phase was detected. The chiral aggregation occurred at the N to DC phase transition. The origin of the supramolecular chirality of the trimer in the DC phase was different from that of the flexible dimer in the $N_{TB}$ phase.

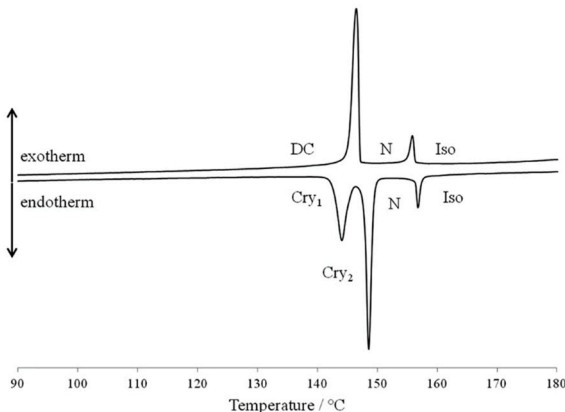

**Figure 16.** DSC thermogram of trimer **I-(7,7)**. The rate of cooling and heating was 5 °C min$^{-1}$. The picture is taken from ref. [68].

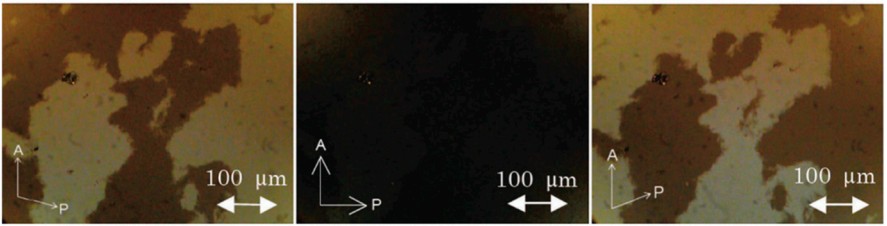

**Figure 17.** Polarized optical textures of trimer **I-(7,7)** on a glass slide with a glass cover in the DC phase at 130 °C under uncrossed and crossed polarizers. The textures are taken from ref. [68].

X-ray diffraction measurements at $T - T_{\text{N-DC}} = -13$ K revealed that the DC phase has a layer structure with a periodicity of 56.5 Å and that the correlation length for the periodicity is 350 Å, which corresponds to about six layers. The extended molecular length for trimer **I-(7,7)** with all-trans conformation of the spacers is estimated to be 68 Å (Figure 18a). On the other hand, the molecular length with a twisted conformation is estimated as 57 Å (Figure 18b).

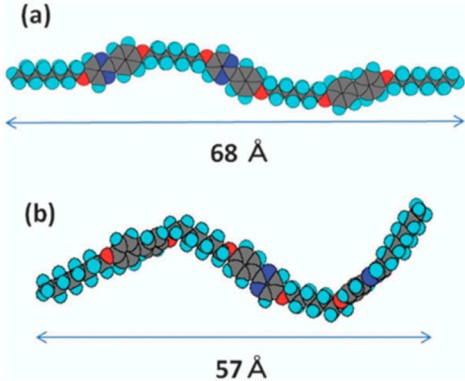

**Figure 18.** MOPAC models for trimer **I-(7,7)** with (**a**) an extended conformation and (**b**) a twisted conformation. The images are taken from ref. [68].

As shown in Figure 15, the trimers with longer and odd-numbered spacers show the DC phase. If we assume that spacers of trimers form an all-trans conformation, a trimer with odd-numbered spacers has a zigzag shape in which all three mesogenic units are inclined with respect to each other, whereas a trimer with even-numbered spacers has a linear shape in which the three mesogenic units are co-parallel. Generally, the flexibility of a liquid crystal trimer increases with the increasing spacer length. Therefore, coupling between the zigzag shape and flexibility is thought to play an important role in the appearance of the DC phase. We explain the origin of the supramolecular chirality as follows. The trimer forms an achiral conformer in the nematic phase, but via intermolecular core-core interactions, it adopts a metastable twisted conformer and begins to exhibit the spontaneous mirror symmetry breaking in the low-temperature DC phase. A schematic model for the N–DC phase transition of the trimer is portrayed in Figure 19 [68]. The achiral trimer produces a dynamic chiral origin.

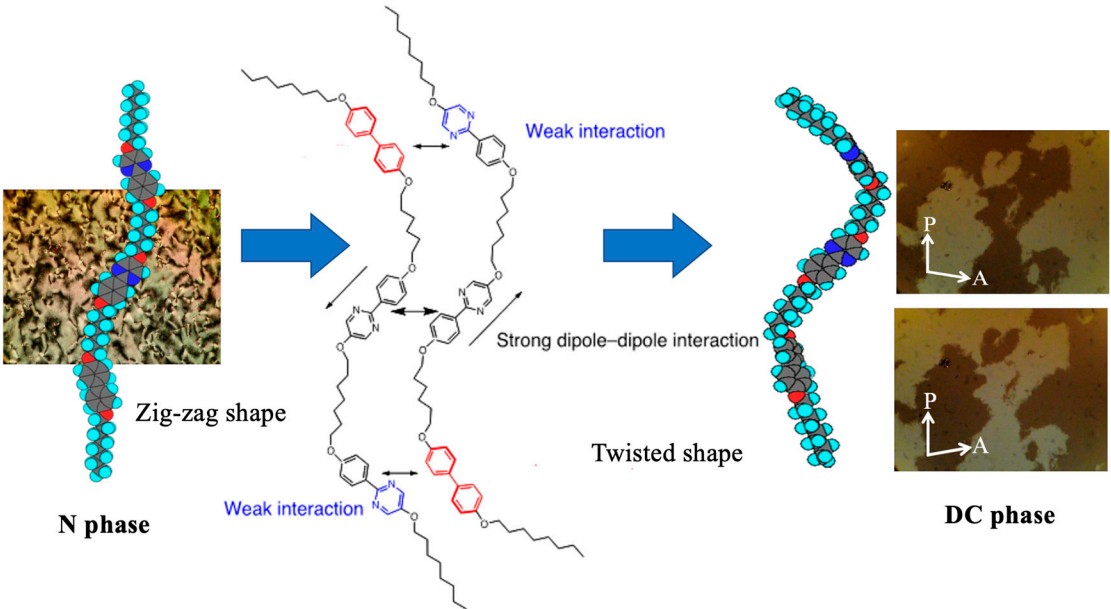

**Figure 19.** Schematic model for the N–DC phase transition of trimer **I-(7,7)**.

Critical differences in physical properties exist between the DC phase of the flexible trimer and the HNF phase composed of rigid bent-core molecules. Figure 20 presents the atomic force microscopy (AFM) images of the surface structures of trimers **I-(7,7)**, **I-(7,9)**, **I-(9,9)**, and **I-(9,11)** in the DC phases at room temperature [69]. The surface structure depends on their spacer length. Trimer **I-(7,7)** showed a disordered surface relief exhibiting dimples with a diameter of $40 \pm 20$ nm, whereas trimer **I-(9,9)** showed a well-ordered surface relief exhibiting dimples with a diameter of $95 \pm 15$ nm.

Figure 21 shows the field-emission scanning electron micrograph (FE-SEM) images of both surface and bulk structures of trimer **I-(9,9)**. Bicontinuous networks are shown at the air/liquid crystal interface (Figure 21a). Although not only well-ordered regions but also some disordered regions exist, the networks accompanying hexagonal voids are organized (Figure 21b). The network consists of bicontinuous triply branchings, and defects exist in the boundaries between adjacent branches. The cross-section area shows a sponge-like structure accompanying voids (Figure 21c). In comparison to lipid–water systems, a sponge phase is essentially a disordered version of the bicontinuous cubic phases [70]. Both a gyroid-like surface and sponge-like bulk are thought to consist of bicontinuous networks, and they are closely related to each other. We explain the mechanism of the formation process as follows. Upon cooling to the DC phase, layer ordering occurs and then the layers are helically deformed at the surface, producing an ordered cubic structure. On the other hand, both the layer formation and the helical ordering of trimer molecules occur

simultaneously in the bulk. The competition between those ordering induces frustration in the system, which can be released via layer deformation to the sponge structure in the bulk [69]. It should be noted that the aggregation of achiral trimers forms the bicontinuous cubic surface with supramolecular chirality.

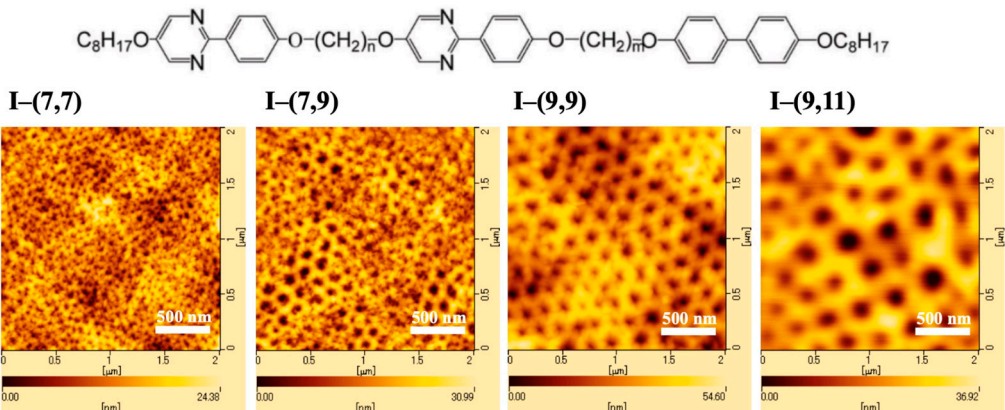

**Figure 20.** AFM images of trimers **I-(7,7)**, **I-(7,9)**, **I-(9,9)**, and **I-(9,11)** in the DC phases at room temperature. Each image shows a microstructure at the air/liquid crystal interface of a droplet sample on an untreated glass substrate. The images are taken from ref. [69].

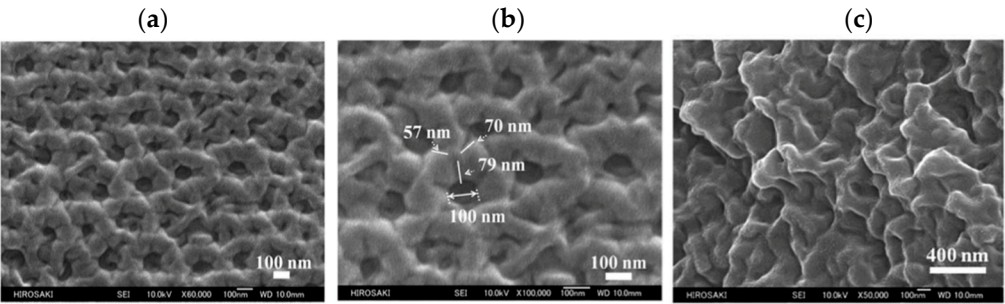

**Figure 21.** (**a**) FE-SEM image of the surface structure of **I-(9,9)**. (**b**) Higher magnification of the surface structure of (**a**,**c**) that of the cross-section area. The picture is taken from ref. [69].

Although the asymmetric trimers **I-(*n*,*m*)** exhibited spontaneous mirror symmetry breaking, there were not only a few domains with opposite handedness that coexisted in the DC phase. Interestingly, symmetric trimers **II-*n*** showed a large homochiral domain [71]. Figure 22 shows the phase transition diagram on cooling for a homolog series of trimers **II-*n***. The trimers possessing even-numbered methylene spacers showed nematic and smectic phases. No chiral property was observed in those phases. On the other hand, all the trimers possessing odd-numbered methylene spacers exhibited a direct Iso–DC phase transition. The super cooling DC phases of trimers **II-*n*** were stable at room temperature similarly to those of trimers **I-*n***. These odd-numbered trimers tend to form homochiral droplets. Figure 23 shows two droplets with opposite handedness of timer **II-9** in the DC phase. The diameter of the homochiral domain was about 1 mm. According to the XRD measurements, trimer **II-9** has a layer structure with a periodicity length of 43.1 Å. The correlation length was about 700 Å. Its molecular length is estimated to be 67 Å. Therefore, the trimer is thought to have an intercalated layer structure, as shown in Figure 24. XRD studies indicate that the other odd-numbered trimers also have an intercalated layer structure.

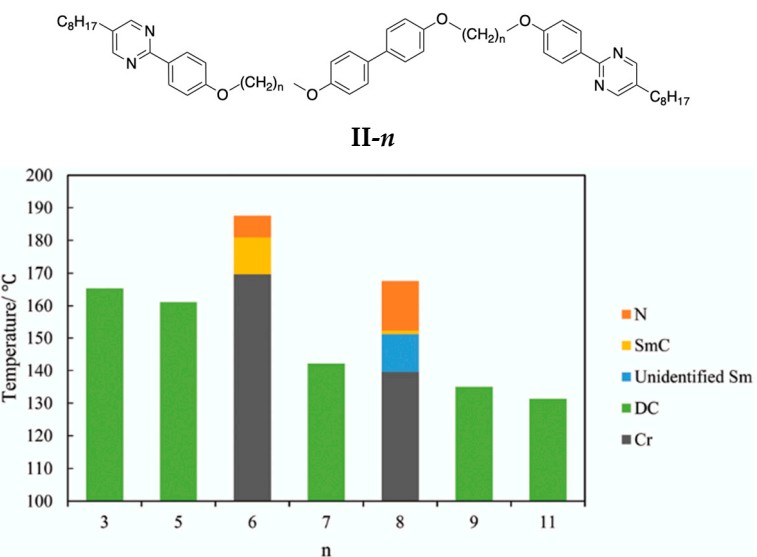

**II-*n***

**Figure 22.** Phase transition diagram on the cooling of trimers **II-*n***. The picture is taken from ref. [71].

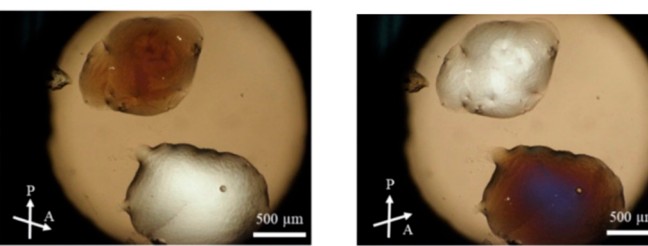

**Figure 23.** Polarized optical textures of trimer **II-9** in the DC phase under uncrossed polarizers. The textures are taken from ref. [71].

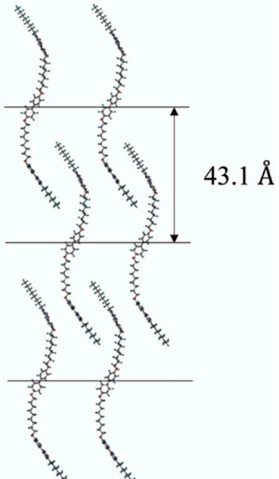

**Figure 24.** Molecular organization model for trimer **II-9** in the DC phase. The picture is taken from ref. [71].

Interestingly, the nanostructures of trimer **II-*n*** depend on the spacer length. Figure 25 shows the FE-SEM images of the surface structures of trimers **II-5** and **II-9**. Trimer **II-5** exhibited hollow cylinders, whereas trimer **II-9** exhibited toroidal pits. Their FE-SEM images of the structures of the cross-section area revealed that both trimers form disordered structures in the bulk material. Furthermore, trimers **II-7** and **II-11** formed toroidal pits at the surface similarly to trimer **II-9**.

(**a**) II-5          (**b**) II-9

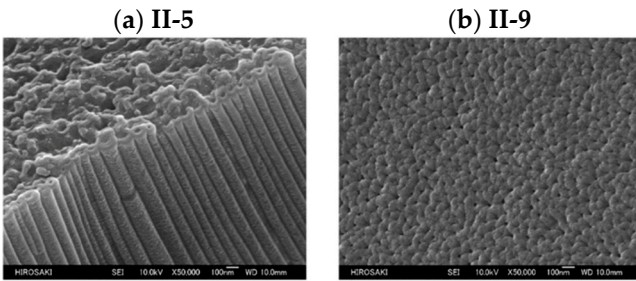

**Figure 25.** (**a**) FE-SEM image of the nanostructure at the air/trimer interface of trimer **II-5** and (**b**) that of trimer **II-9**. The FE-SEM measurements were carried out at room temperature. The images are taken from ref. [71].

Figure 26 presents an FE-SEM image of the nanostructure at the air/trimer interface and in the cross-section area of trimer **II-3**. The trimer formed cylinders at the surface, whereas double-winding cylindrical tubes were formed in the bulk of the material. In the expansion figure, a spiral-like periodicity can be seen for the tube.

(**a**)          (**b**)          (**c**)

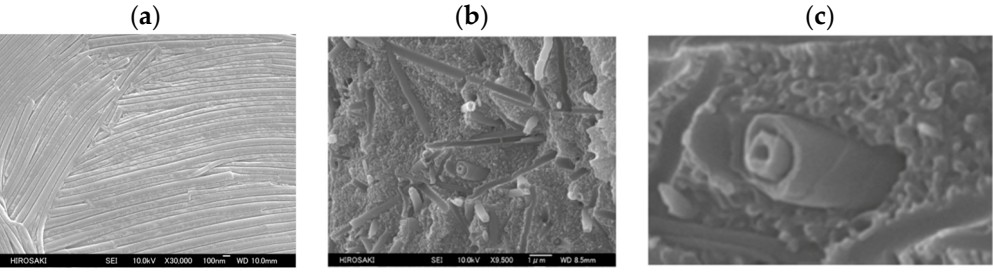

**Figure 26.** (**a**) FE-SEM image of the microstructure at the surface of trimer **II-3**, (**b**) that of the cross-section area, and (**c**) an enlarged image of the central part of (**b**) for clarity. The SEM measurements were carried out at room temperature. The images are taken from ref. [71].

The spacer effect on the formation of microstructures of the trimers possessing odd-numbered spacers is unexpectable. Figure 27a shows an FE-SEM image of an equimolecular mixture of trimer **II-7** and **II-11**. The mixture exhibited toroidal pits at the surface, indicating that their driving forces to form the nanostructure are the same. On the other hand, rounded tubes and toroidal pits coexisted at the surfaces of **II-5** and **II-7**, as shown in Figure 27b. Their driving forces for the formation of their DC phase are different. Figure 27c shows an FE-SEM image of the surface structure of trimers **II-3** and **II-5**. The mixture formed cylindrical tubes. The mixture exhibited chiral conglomerates, and each homochiral domain size was about several mm². However, the DC phase of trimer **II-5** was not miscible with that of trimer **II-3**, suggesting that their driving forces for the DC phase formation are different.

(**a**)          (**b**)          (**c**)

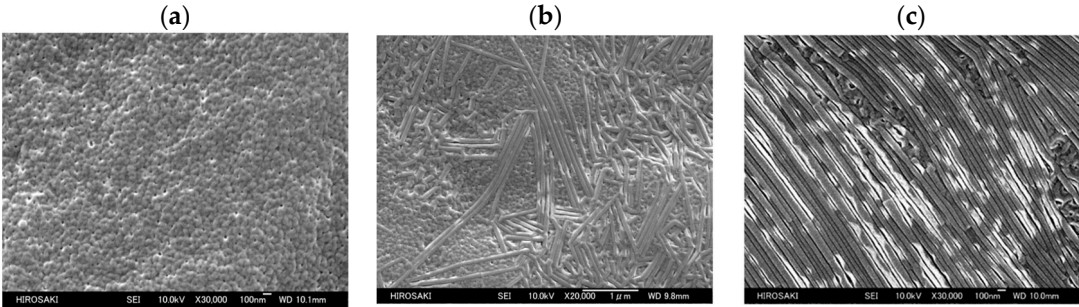

**Figure 27.** (**a**) FE-SEM image of the microstructure at the surface of an equimolecular mixture of trimers **II-7** and **II-11**, (**b**) that of trimers **II-5** and **II-7**, and (**c**) that of trimers **II-3** and **II-5**. The images are taken from ref. [71].

The present symmetric trimers **II-*n*** exhibited diverse supramolecular architectures from cylindrical tubes to toroidal pits depending on the spacer length in their DC phases. We discuss the formation of the supramolecular architectures using a molecular organization model (Figure 28). Shorter spacer trimers **II-3** and **II-5** induce their helical axes parallel to the normal layer to form cylindrical tubes. On the other hand, longer spacer trimers **II-7**, **II-9**, and **II-11** induce their helical axes perpendicular to the normal layer. The formation of diverse supramolecular architectures consisting of the trimer molecules can be explained in terms of the direction of chirality synchronization.

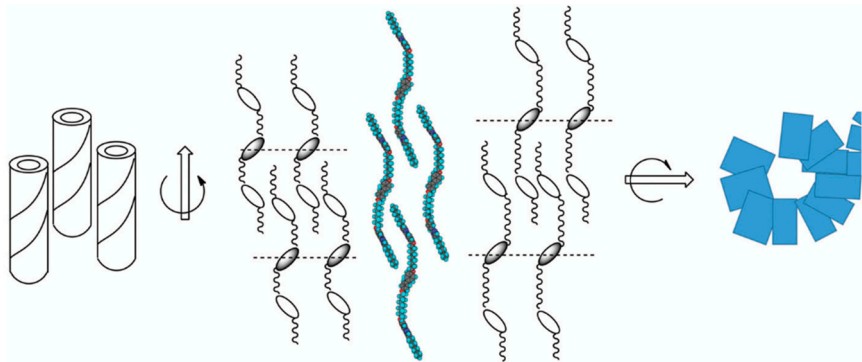

**Figure 28.** Molecular organization model for the formation of the supramolecular architectures. The arrows show their helical axes. The picture is taken from ref. [71].

## 4. External-Field-Induced Mirror Symmetry Breaking in Liquid Crystals

The photo-induced isomerization of an azobenzene unit is well known to cause a remarkable change in the physical properties of liquid crystals. Azobenzene derivatives are expected to be useful for applications in photomemory [72,73], photoalignment [74–79], or photomobile [80,81]. A lot of research on photoinduced chirality modulation has been reported. For example, Tamaoki and Wada reported newly synthesized bicyclic azobenzene dimers that possess enantiomers whose racemization rates could be reversibly controlled via E–Z photoisomerization of the azobenzene units [82]. Upon alternating the exposure to *r*- and *l*-CPL, they were able to repeatedly perform partial enrichment of (S)- and (R)-enantiomers, respectively [82]. Serrano et al. reported that the columnar mesomorphic assembles of propeller-like hydrogen-bonded complexes showed a chiral photoresponse upon irradiation with CPL of a given handedness [83]. Iftime reported that irradiation with the circularly polarized light of a film of an achiral azobenzene liquid-crystalline polymer induces chirality [84]. Recently, photo-induced isothermal phase transitions in chiral conglomerates composed of achiral liquid-crystalline molecules have been observed. Some examples are shown in Figure 29 [85–88]. Almost all the photo-induced isothermal phase transitions are chiral to achiral phase transitions, except for the SmA to $Iso_1^*$ transition [87].

Although the photo-induced *cis*-azobenzene unit is well known to contribute to the production of chirality at a molecular level, the correlation of a *cis*-isomer and the chiral conformation still remains a problem. According to a MOPAC model of a *cis*-azobenzene unit, it can form a twist–bent structure. To the best of our knowledge, there is no evident proof that a *cis*-azobenzene unit performs as a chiral origin. If we design an achiral azobenzene derivative that intercalates to the adjacent layers composed of host smectic molecules, the photo-induced twist–bent structure of the guest molecule can induce chirality in the smectic phase due to the guest–host interactions in comparison to the intercalated chirality model (see Figure 10). We designed azobenzene trimer **III-*n*** (*n* = 7–11) and used a guest-host system of the azobenzene trimer and a host LC [89]. Their molecular structures of trimer **III-*n*** and **Host LC** are shown in Figure 30. Almost all the trimers except trimer **III-7** showed a nematic phase. The host LC (**Host LC**) exhibited N and smectic C (SmC) phases with wide temperatures.

**HNF to Iso**

**N$_{TB}$ to N**

**SmA to chiral isotropic (Iso$_1$\*)**

**DC to N**

**Figure 29.** Achiral molecules exhibiting photo-induced isothermal phase transitions. HNF to Iso [85]; N$_{TB}$ to N [86]; SmA to Iso$_1$\* [87]; DC to N [88].

**III-*n***

**Host LC**

**Figure 30.** Molecular structures of trimer **III-*n*** and **Host LC** [89].

The phase transition behavior of a mixture of trimer **III-11** (20 wt%) and **Host LC** (80 wt%) on a glass slide with a glass cover with 365 nm UV irradiation at a power of 20 mW cm$^{-2}$ was as follows. On cooling from the isotropic liquid, the N phase and then the SmC phase appeared in which the molecules were aligned homeotropically. The SmC phase exhibited chiral conglomerates with bright grain boundaries as shown in Figure 31. A mixture of trimer **III-9** (20 wt%) and **Host LC** (80 wt%) showed a similar transition behavior as a mixture of trimer **III-11** and **Host LC**. In the case of trimer **III-10** possessing even-numbered spacers, its mixture with **Host LC** showed chiral conglomerates without grain boundaries (Figure 32). With respect to trimers **III-7** and **III-8**, the mixtures with **Host LC** exhibited neither grain boundaries nor chiral conglomerates.

The trimers themselves did not exhibit a chiral nature in their nematic phase by photoirradiation. Why did timers **III-9**, **III-10**, and **III-11** show supramolecular chirality? We explain the appearance according to the intercalated chirality model (see Figure 10). The photo-induced *cis*-isomer of trimer **III-*n*** (*n* = 9, 10, and 11) intercalating with the adjacent layers produces guest–host interactions in each layer. The twist–bend form of the *cis*-azobenzene unit is synchronized via a cooperative motion of the host core parts to produce the chiral conglomerates. With respect to trimers **III-7** and **III-8**, they cannot sufficiently intercalate with the adjacent layer due to their sorter spacers. Figure 33a shows a model for the formation of the helical layer structure of a mixture of *cis*-isomer **III-10** and

**Host LC**. The director is twisted almost continuously (see the red arrows in Figure 33a). On the other hand, grain boundaries were observed only for odd-numbered trimers **III-9** and **III-11**. We surmise that this phenomenon is attributed to the marked difference in overall molecular structure between trimers **III-10** and **III-11**. Their MOPAC models are shown in Figure 34. Figure 33b shows a schematic model for the formation of the helical layer structure accompanying the bend deformation of the director. The chiral conglomerates are formed as explained above. The zigzag molecular shape of the odd-numbered trimer might induce bend layer deformation in the smectic phase of **Host LC**. Generally, a bend deformation in a smectic phase is forbidden because it prohibits the formation of a layer structure with a constant layer spacing. The competition between the bend deformation and the desire to form a layer structure causes grain boundaries between smectic blocks. The photoinduced chirality at a molecular level is amplified via guest-host interactions to produce the supramolecular chirality. We can say that a *cis*-azobenzene unit with a twist–bent structure potentially acts as the origin of chirality at a molecular level.

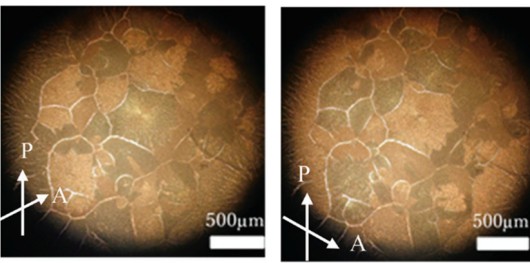

**Figure 31.** Optical textures of a mixture of trimer **III-11** (20 wt%) and **Host LC** (80 wt%) cooling from the isotropic liquid with 365 nm UV irradiation at a power of 20 mW cm$^{-2}$ between uncrossed polarizers. The textures are taken from ref. [89].

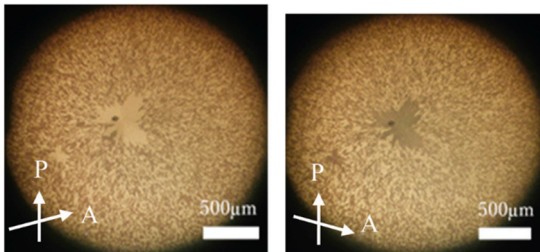

**Figure 32.** Optical textures of a mixture of trimer **III-8** (20 wt%) and **Host LC** (80 wt%) cooling from the isotropic liquid under the same condition. The textures are taken from ref. [89].

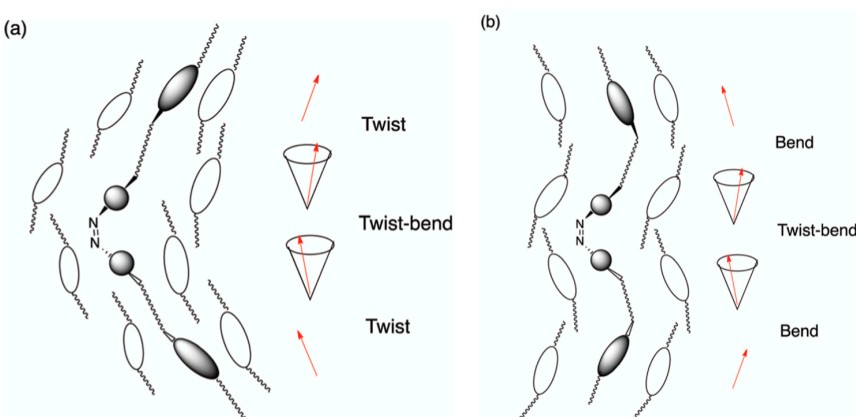

**Figure 33.** (**a**) Schematic model for the formation of chiral conglomerates for a mixture of photo-induced *cis*-isomer **III-10** (20 wt%) and **Host LC** (80 wt%). (**b**) Model for the formation of chiral conglomerates accompanying grain boundaries for a mixture of photo-induced *cis*-isomer **III-11** (20 wt%) and **Host LC**. (80 wt%). The pictures are from ref. [89].

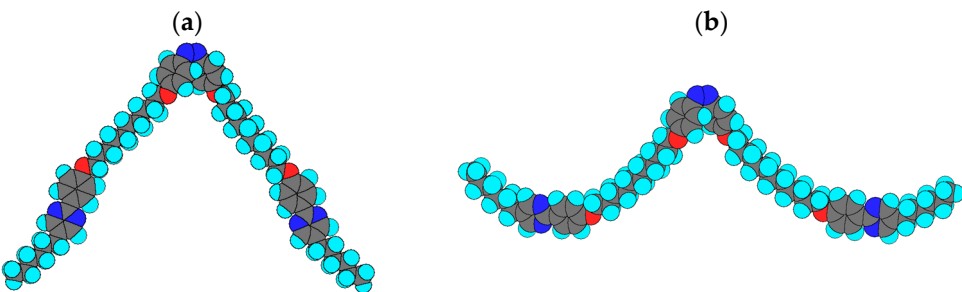

**Figure 34.** MOPAC models for *cis*-isomers of trimers **III-10** and **III-11**. The pictures are from ref. [86].
(**a**) III-10, (**b**) III-11.

The electric field can also induce mirror symmetry breaking in liquid crystals. However, there are a few reports about electric-field-induced helix formation in an LC phase of achiral molecules [90,91]. One of the reports is a study on a bent-core LC under an AC field. When an AC field was applied on the bent-core LC in the synclinic and antiferroelectric $SmC_sP_A$ phase and then the field was switched off, the heliconical $Sm(CP)^{hel}$ phase was found to be induced as shown in Figure 35 [90]. The bent-core molecules are reoriented in each layer by an electric field. The molecules have a rigid twist conformation. The synchronized helical conformers prefer a correlation of the tilt angle between the layers that are unequal to 0° (synclinic) or 180° (anticlinic), thus leading to an intermediate angle and providing the twist between the layers.

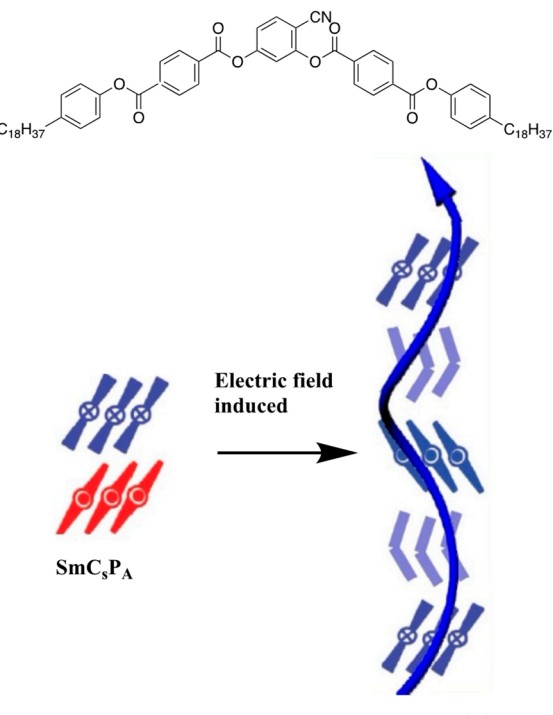

**Figure 35.** Model for the electric-field-induced $SmC_sP_A$ to $Sm(CP)^{hel}$ transition. The picture is taken from ref. [90].

The other report is a study on an achiral H-shaped trimer as shown in Figure 36 [91]. The trimer exhibited a monotropic nematic phase. When applying an AC field of 6.0 V $\mu m^{-1}$ on the H-shaped trimer, the texture was completely dark, revealing that the molecules were aligned perpendicular to the substrate. Upon removing the electric field, the dark color changed to orange (Figure 37).

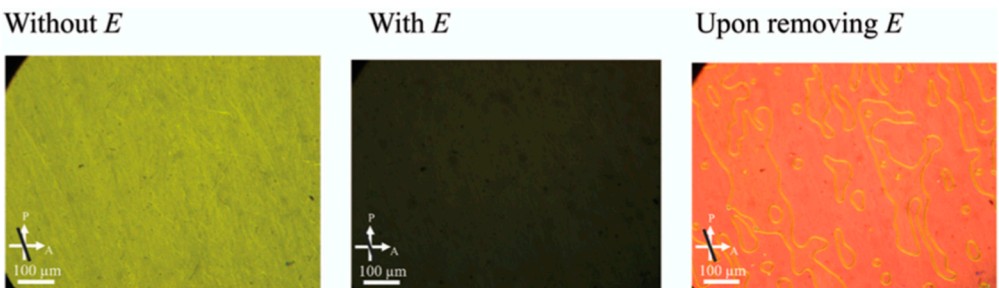

**Figure 36.** Molecular structures of an achiral H-shaped trimer and its corresponding monomeric rod-like LC [91].

**Figure 37.** Electric-field-induced textures of the trimer with $E = \pm 6.0$ Vμm$^{-1}$. The sample was confined in a 5 mm homogeneously aligned cell. The black bars indicate the rubbing direction. The textures are taken from ref. [91].

We investigated the dynamics of reorientation of the molecules upon removing the electric field. Before applying the electric field, the texture was dark, revealing that the molecules were aligned uniformly parallel to the rubbing direction in the N phase (Figure 38a). When applying an AC field of 12 Vμm$^{-1}$, the texture was almost dark. Immediately after the removal of the electric field, a nematic texture appeared (Figure 38b). However, the dark texture as shown in Figure 38a was not obtained. The molecules are not aligned uniformly. The light green color changed to orange (Figure 38d) through the mottled texture consisting of gray and dark domains (Figure 38c). By observing the sample between slightly uncrossed polarizers, the orange texture was split into red and dark yellow domains (Figure 38e,f). They have optical activity with opposite handedness. With respect to the monomeric rod-like LC, upon removing the electric field, an N*-like texture appeared during the reorientation process, and it changed to the original N texture in 1 s. The rod-like molecules reorient once again parallel to the substrate uniformly.

Figure 39 shows the optical textures at the equilibrium state after the removal of the electric field using a 10 μm cell. A large homochiral domain can be seen. The molecular alignment force in a 10 μm cell is thought to be weaker than that in a 5 μm cell. The molecular alignment force is a necessary requirement for the appearance of the electric-field-induced N* phase. The optimization of the experimental conditions can lead to supramolecular homochirality.

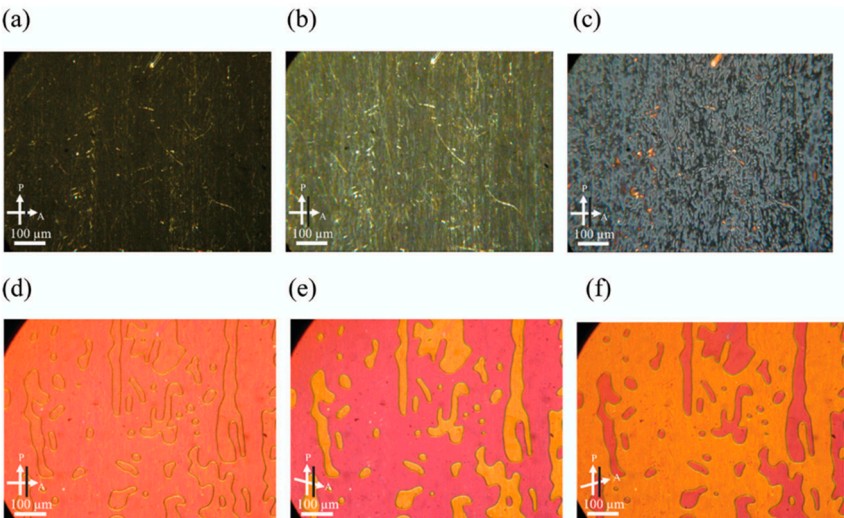

**Figure 38.** Optical textures of the trimer after the removal of an AC field of 12 V μm$^{-1}$. The sample was confined in a 5 μm homogeneously aligned cell. (**a**) Texture before applying the electric field. (**b**) Texture at 2.5 s after the removal of the electric field. (**c**) Texture at 11 s after the removal of the electric field. (**d**–**f**) Textures at 17 s after the removal of the electric field between crossed and uncrossed polarizers. The textures are taken from ref. [91].

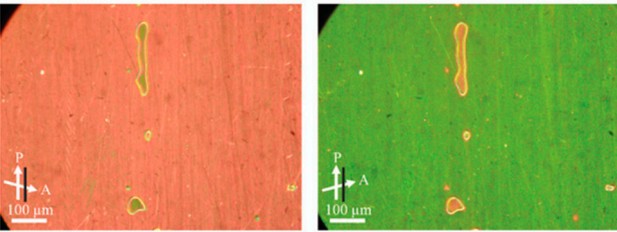

**Figure 39.** Optical textures of the trimer after the removal of an AC field of 12 V μm$^{-1}$ between crossed and uncrossed polarizers. The compound was confined in a 10 μm homogeneously aligned cell. The textures are taken from ref. [91].

Let us discuss how the supramolecular chirality appears using a schematic model (Figure 40) [91]. The H-shaped trimer forms a rod-like shape **A** keeping the alkyl chains outside without the electric field (Figure 40a). The director is parallel to the rubbing direction. Applying an AC field above the threshold voltage on the sample, the director reorients perpendicular to the substrate. During the process, the molecules accompany the conformational change from **A** to **B** as shown in Figure 40b. The molecules form a rod-like shape **B** turning the terminal cyano units toward the top and bottom electrodes. Upon removing the electric field, the twisted shape **C** is thought to be formed during the reorganization process. The metastable twisted molecules can exist in bulk, whereas the molecules forming rod-like shape **A** orient parallel to the rubbing direction near the surface (Figure 40c). The twisted shape **C** returns to the rod-like shape **A** to drive helical interactions among molecules, producing a helical molecular alignment (Figure 40d). The coupling of the on/off switching of the electric field and the surface anchoring is responsible for breaking the mirror symmetry in the nematic phase of the flexible H-shaped trimer.

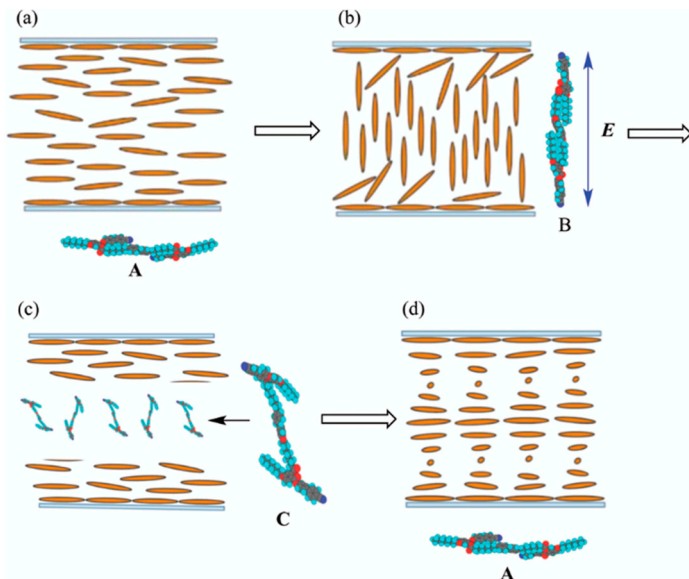

**Figure 40.** Model for the electric-field-induced phase transition. (**a**) Molecular orientaion without an AC field, (**b**) molecular reorientation with an AC field, (**c**) that upon removing the AC field, and (**d**) molecular orientation producing the helical alignment. The picture is taken from ref. [91].

## 5. The Formation of Helical Polymers without a Chiral Component

A helical polymer network composed of achiral molecules, which is derived from a blue phase (BP), can be a template for the formation of a helical polymer. BPs have a frustrated structure which is stabilized by the chirality-dependent defects [92,93]. They are classified into three categories depending on the cylinder packing structure: blue phase I (BPI), blue phase II (BPII), and blue phase III (BPIII). BPI and BPII have a cubic structure, while BPIII has an amorphous one (Figure 41). Within each cylinder, molecules are twisted from −45° at one end to +45° at the other hand.

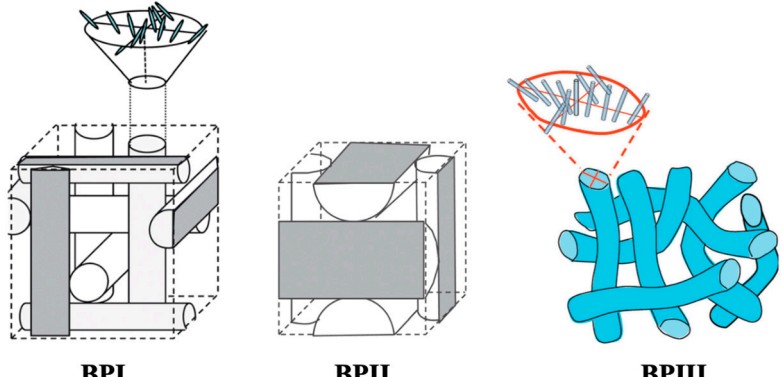

**Figure 41.** Schematic representations of the blue phase structures. The pictures of BPI and BPII are taken from ref. [94]. The picture of BPIII is taken from ref. [95].

Usually, blue phases are found in an extremely narrow temperature range (ca. 1K). Kikuchi et al. reported that specific polymer networks can stabilize the lattice defects of BPI [96]. Coles et al. demonstrated the fabrication of self-assembled three-dimensional nanostructures by polymer templating BPI, as portrayed in Figure 42 [97]. Functional monomers were photopolymerized in the BPI composed of chiral materials. Removing the low-molecular-mass materials created a porous cast. By refilling the cast with an achiral nematic liquid crystal, templated blue phases with wide temperature ranges appeared.

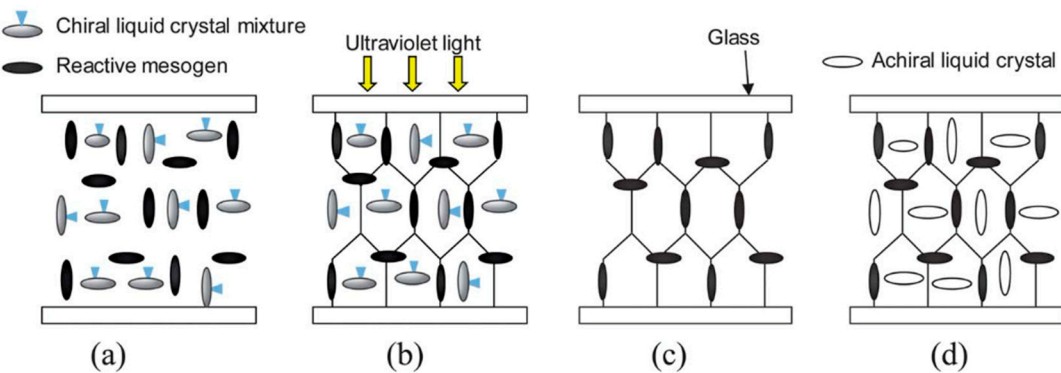

**Figure 42.** Schematic diagram of the formation of the 3D nanostructured polymer [97]. (**a**) Chiral LC mixture forms a blue phase. (**b**) Cell is exposed to UV light to photopolymerize the reactive mesogens. (**c**) Removing the liquid crystal, chiral dopant, and remaining reactive mesogen–photoinitiator mixture creates a porous cast. (**d**) An unopened cell is refilled with the achiral nematic liquid crystal. The picture is taken from ref. [94].

Gandhi et al. reported the fabrication of a porous polymer scaffold that mimics the amorphous structure of BPIII at a nanoscale by imprinting a reactive mesogenic polymer network along topological defects in BPIII [98]. Refilling the scaffold with an achiral nematic LC shows a texture and reflection spectrum that is very similar to the original polymer stabilized BPIII (PS-BPIII). Furthermore, the nematic LC-refilled scaffold exhibited comparable transmittance electric field curves and response times to those of the original PS-BPIII. These observations suggest that the polymer nanostructure of the BPIII scaffold memorizes the chiral nature of the local molecular arrangement of the original PS-BPIII and induces optical activity or chirality in the refilled achiral nematic LCs [98]. The authors noted that polymer scaffolds nanoengineered from BPIII may also be used as chiral-induction agents.

Araoka and Choi et al. prepared a polymer network of coexisting nano-pores derived from an HNF phase of bent-core molecules [99]. Figure 43a shows the FE-SEM image. The polymer network composed of achiral molecules exhibited a CD response, revealing that it has a chiral superstructure. Scanning electron microscopy confirmed the formation of an inverse nanohelical structure via photo-cross-linking. The schematic image of the polymer network and removal of HNF is shown in Figure 43b. Upon refilling the polymer network with achiral nematic LC molecules, a chiral conglomerate phase appeared due to the transfer of chirality from the inverse helical network to achiral nematic LC molecules. Figure 44 shows its POM textures. The proposed method based on bottom-up templating for obtaining chiral structures without a chiral component is very interesting. However, HNF phases are usually exhibited as chiral conglomerates. Therefore, it is difficult to obtain a homochiral template using this system.

Recently, we reported a chirality transfer system using a blue phase as a three-dimensional template for the helical polymerization of achiral monomers [100]. We used a BP mixture of a **T-shaped LC**, **S-** or **R-811**, a reactive monomer mixture (**RM257** and **C12A**), and a photo-initiator **DMAP**. Their molecular structures are shown in Figure 45. It exhibited a phase sequence of Iso–BPIII–cubic BP. The polymer stabilization was performed in the BPIII. Figure 46 shows the CD spectra of the BPIII polymer network derived from the BP mixture containing **S-811** or **R-811**. Both networks exhibited a strong CD signal at about 305 nm and their optical senses were opposite. Several weak but clear signals can be seen in the wavelength region below 300 nm. They are mirror images. Those peaks are thought to have originated from the molecular twist of the mesogenic core of bifunctional monomer **RM257**. Figure 47 shows a schematic model of the polymer network and its FE-SEM image. The chiral field originated from the twisting power in the BPIII induces the axial chirality of the mesogenic core of **RM257**. As the photopolymerization progresses,

the induced chirality propagates to form a homochiral polymer network. It produces the supramolecular chirality in the BPIII network.

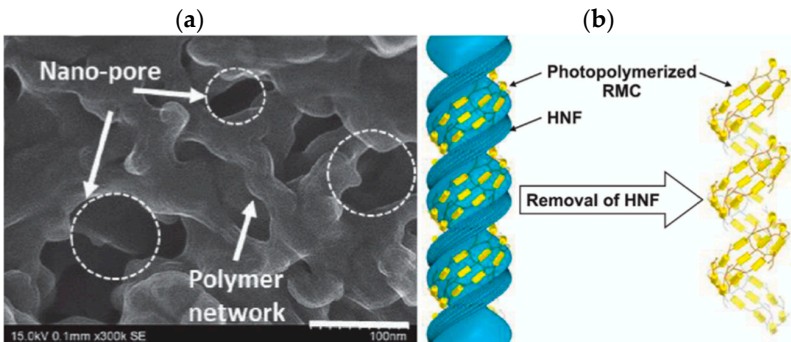

**Figure 43.** (**a**) FE-SEM image of the polymer network coexisting nano-pores derived from a helical nanofilament phase. (**b**) Schematic image of the polymer network and removal of HNF. Reprinted with permission from ref. [99]. Copyright 2020 American Chemical Society.

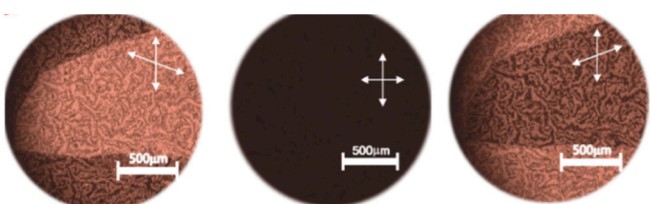

**Figure 44.** Polarized optical textures of the nanoporous film filling with an achiral nematic LC mixture under crossed and uncrossed polarizers. Reprinted with permission from ref. [99]. Copyright 2020 American Chemical Society.

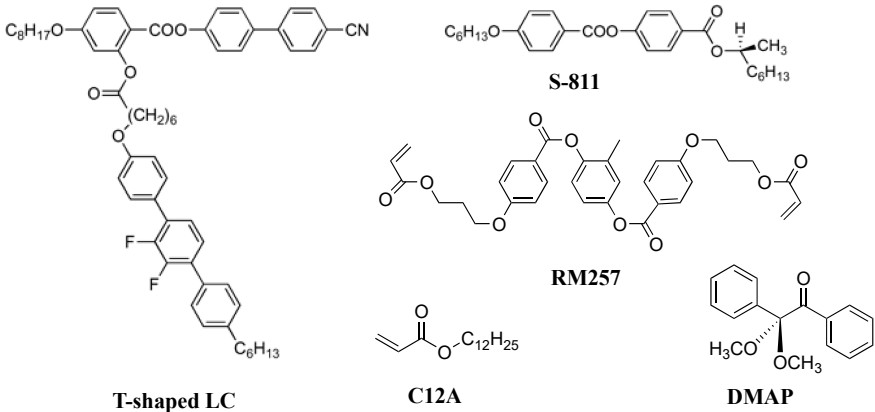

**Figure 45.** Molecular structures of the BP mixture [100].

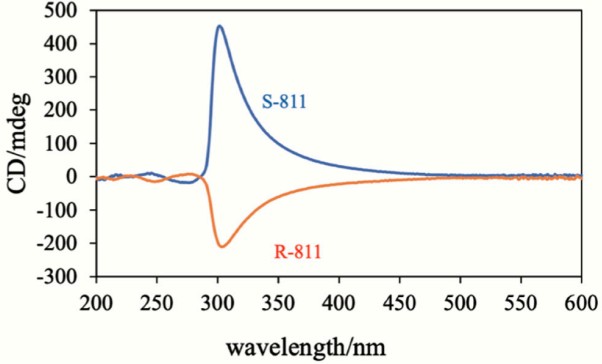

**Figure 46.** CD spectra of the BPIII polymer network derived from the BP mixture containing **S-811** or **R-811**. Reprinted with permission from ref. [100]. Copyright 2023 Taylor and Francis.

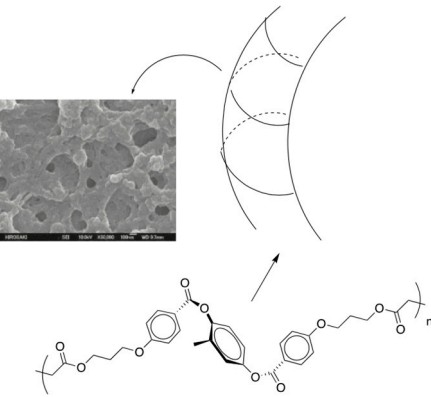

**Figure 47.** Schematic model of the polymer network and its FE-SEM image. Reprinted with permission from ref. [100]. Copyright 2023 Taylor and Francis.

With respect to the photopolymerization of achiral monomers using the BPIII polymer network, we used a mixture of 4-[(6-acryloyloxy)hexyloxy]-4′-cyanobiphenyl (**AHCB**, 14 wt%), 1,1,1-tris(acryloyloxymethyl)propane (**TMPTA**, 81 wt%), and a photo-initiator (5.0 wt%) (Figure 48). The polymer film exhibited a negative CD with strong intensity at around 350 nm as shown in Figure 49. The optical sense of the film was the same as that of the polymer network. According to the SEM image of the polymer network, it has a porous structure, and its pore sizes are about 100–200 nm. Therefore, it is realistic that the reactive monomers gather in the pores, and they are photopolymerized along the pores. The supramolecular chirality of the BP network is transferred to the corresponding polymer film during the photopolymerization without inducing a microscopic molecular chirality. The transfer system can produce the helical polymers from desired achiral monomers.

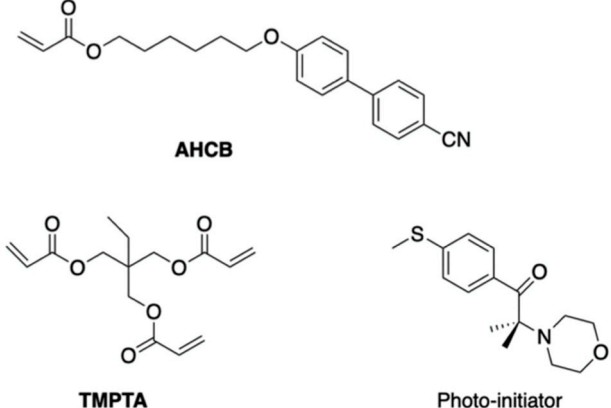

**Figure 48.** Molecular structures of **AHCB**, **TMPTA**, and a photo-initiator [100].

A chiral component is necessary for using the BP template. If we use the supramolecular chirality in the DC phase consisting of achiral trimer molecules instead of such a chiral component, we can prepare a chiral material without a chiral source [101]. As described above, the supramolecular chiral phases organized by symmetric trimers **II-*n*** exist as chiral conglomerates, where the two chiral domains coexist in a non-equal population [71]. If the predominant domain can be amplified to the whole area, we can obtain a homochiral phase. Choi and Takezoe et al. reported that a large enantiomeric excess in a liquid–crystal phase can be induced by CPL as an external stimulus and that an imbalance in the two chiral domains becomes remarkable in the $B_X$ phase [102]. This system needs an external chiral origin. Although we could not obtain a wide homochiral area from trimer **II-9** itself even after thermal annealing, we succeeded in obtaining a homochiral sample with a suitable area (>1 cm$^2$) for a template reaction by using the following procedure [101]. A mixture of trimer **II-9** (70.6 wt%), **RM257** (23.3 wt%), a photo-initiator, and a thermal polymerization

inhibitor was heated to the isotropic liquid at 155 °C. Then, it was cooled from the isotropic liquid to the DC phase. The DC phase still consisted of several domains with opposite handedness (Figure 50a). It was exposed to UV irradiation. It was heated again to the isotropic liquid at 200 °C and then cooled to 25 °C. Interestingly, the sample existed as a monodomain with homochirality (Figure 50b). The handedness was stochastic. Thus, we call the obtained DC phase a polymer-stabilized DC (PS-DC) phase. The CD spectra of the PS-DC materials with opposite handedness are shown in Figure 51. Very weak signals can be seen in the wavelength region below 399 nm. Those peaks are thought to have originated from the molecular twist of the mesogenic core of bifunctional monomer **RM257**. The predominant domain was amplified to the whole area, providing a homochiral state.

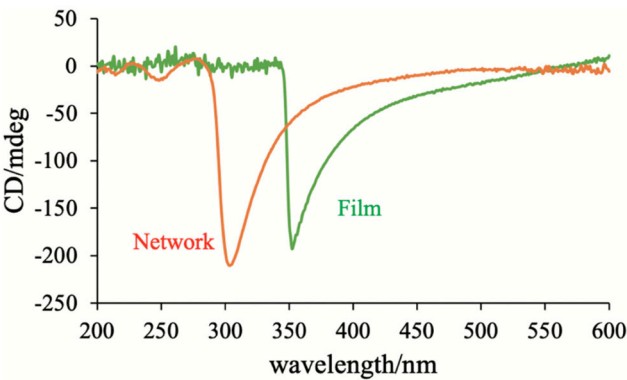

**Figure 49.** CD spectrum of the BPIII polymer network (red) and that of the polymer film (green). Reprinted with permission from ref. [100]. Copyright 2023 Taylor and Francis.

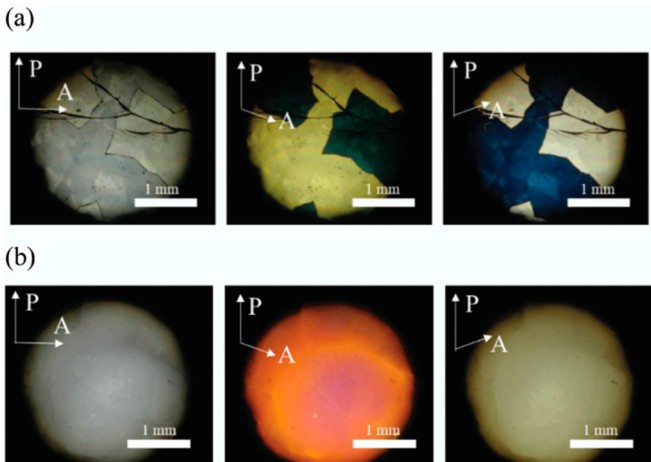

**Figure 50.** (**a**) Optical textures of trimer **II-9** in the DC phase and (**b**) those of the mixture containing trimer **II-9** in the polymer-stabilized DC phase between crossed and uncrossed polarizers at room temperature. The pictures are taken from ref. [101].

In order to understand the amplification mechanism, we compared the time-dependent optical texture in the presence of the polymer network with that in the absence of the polymer network. The movies can be seen in Video S1 [103] (see Supplementary Materials). In the absence of the polymer network, a few domains with opposite handedness appeared and they were spread independently. Clear boundaries existed between two domains with opposite handedness. On the other hand, in the presence of the polymer network, many small homochiral domains appeared and they were spread simultaneously. The spread speed of the latter was much faster than the former. We speculate the formation mechanism of the homochiral PS-DC as follows. The chiral field originating from the twisting power in the DC phase induces transient axial chirality of the **RM257** unit in the polymer network. The twist units with opposite handedness coexist in a non-equal population in the polymer.

Then, chiral amplification by 'Majority rule' produces almost a single helical structure in the polymer network. The rule is that even a small excess of majority units could lead to a preference for epimerization of the minority units to the configuration of the majority units, which would lead to a fuller excess of the majority helical sense [41]. At the Iso–DC transition on cooling from the isotropic liquid, the trimer molecules in contact with thus produced helical polymer network may form many small domains with the same twist sense as the polymer network. They are spread to the whole area rapidly.

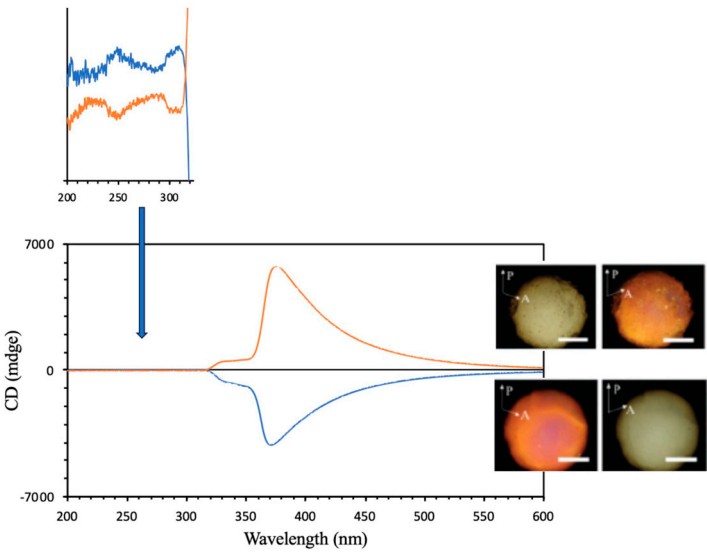

**Figure 51.** CD spectra of the PS-DC materials with opposite handedness and their optical textures under uncrossed polarizers [101].

We performed the photopolymerization of achiral reactive monomers on the surface of the PS-DC material [101]. We used a mixture of benzyl methacrylate (**BM**, 47.5 wt%), 1,1,1-tris(acryloyloxymethyl)propane (**TMPTA**, 47.5 wt%), and a photo-initiator (5.0 wt%). Figure 52 shows a schematic illustration of the preparation process of the photopolymerized film. A transparent polymer film was obtained (also see Figure 52). A comparison of the FE-SEM images between the PS-DC material and the polymer film reveals that the surface structure of the PS-DC material is reversely printed to the surface of the polymer film (Figure 52).

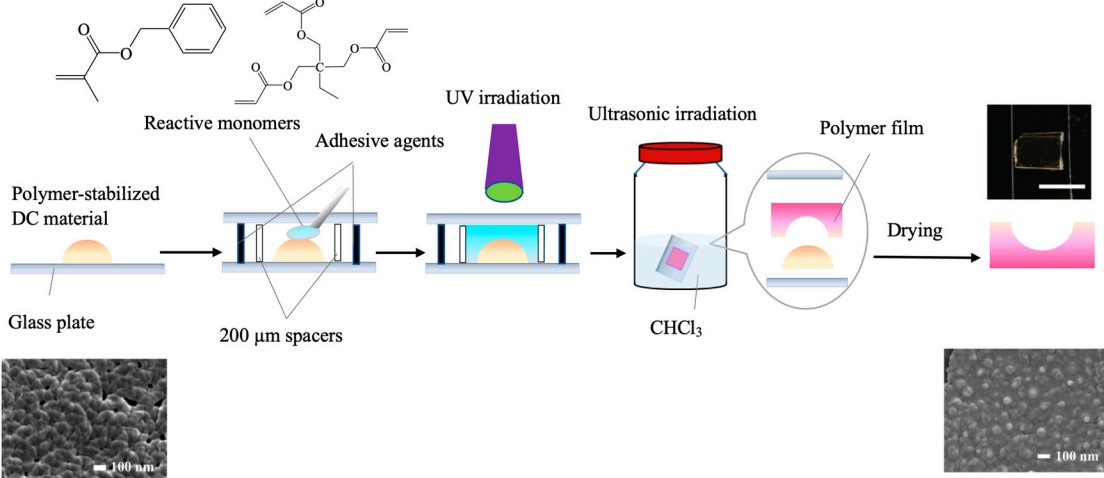

**Figure 52.** Schematic illustration of the preparation process of the photopolymerized film with a photograph of the polymer film on a glass plate and FE-SEM images of the surface structures of the PS-DC material and the polymer film [101].

Figure 53 shows the CD spectra of the polymer films prepared by using each PS-DC material possessing opposite handedness. Each polymer forms a helical structure with an opposite handedness. The supramolecular chirality is transferred from each PS-DC material to the corresponding polymer film. Figure 54 shows a schematic sketch of the transfer process. The surface exhibiting voids with a mesoscopic size has a chiral environment. The helical arrangement of the reactive monomers is induced through the host–guest template effects at the interface in the course of photopolymerization similarly to the chirality transfer system using a blue phase as a three-dimensional template.

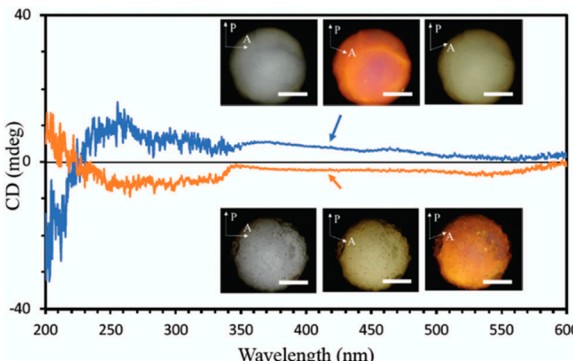

**Figure 53.** CD spectra of the polymer films prepared by using each PS-DC material. Optical textures of the PS-DC materials are inserted. The white bars represent 1 mm. The picture is taken from ref. [101].

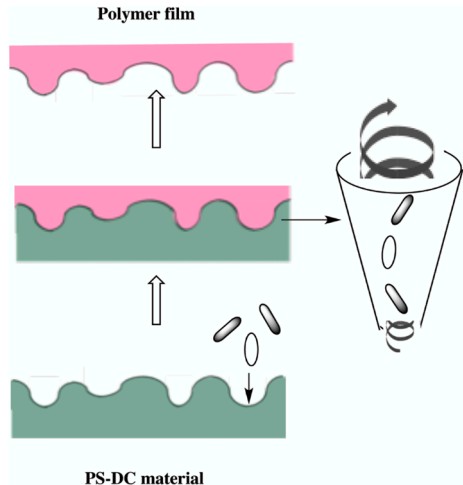

**Figure 54.** Schematic sketch of shape and chirality transfer from the PS-DC material to the polymer film. The picture is taken from ref. [101].

## 6. Concluding Remarks

We show our strategy for producing chiral materials without a chiral component in Figure 55. We describe in this review several liquid crystal trimers producing a dynamic chiral origin. Such trimers are regarded as a supermolecule. These supermolecules are designed by combining static molecular shape and cooperative molecular motion. Spontaneous mirror symmetry breaking in self-assembled achiral trimer molecules induces supramolecular chirality. Chiral domains with opposite handedness exist in non-equal populations. The dominant domain is amplified by a polymer network to produce a homochiral state. Chirality is transferred to the polymer film in the course of polymerization of achiral reactive monomers on the surface.

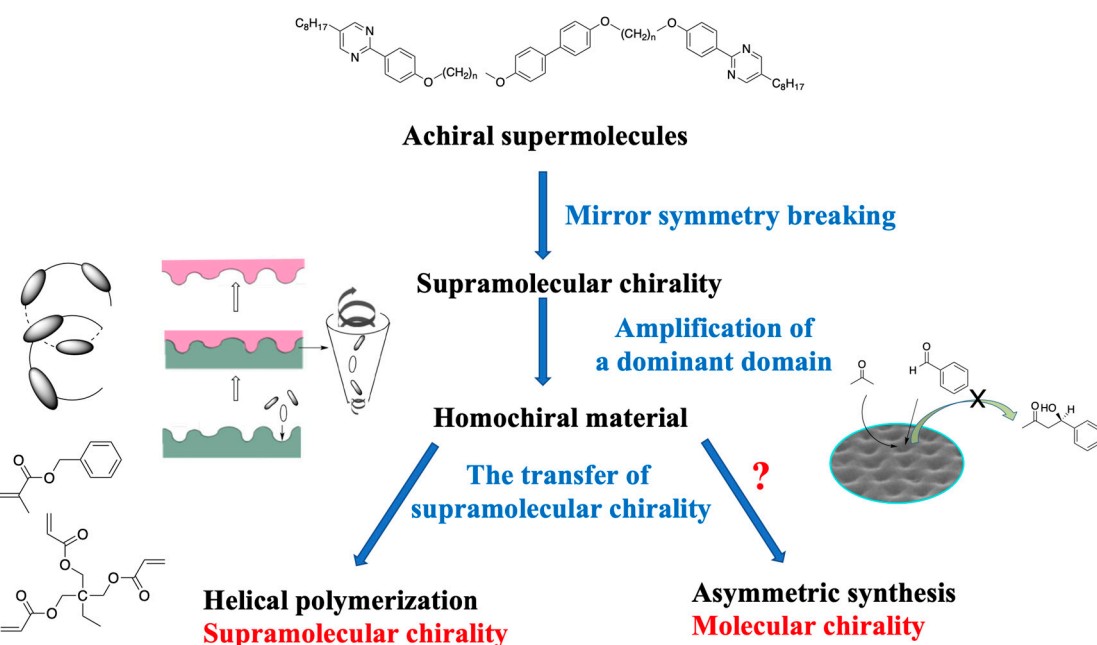

**Figure 55.** Strategy for producing chiral materials without a chiral component.

We preliminary studied the reaction of acetone with benzaldehyde in the presence of a catalytic amount of trimer **I-(9,9)** in the DC phase showing chiral conglomerates [104]. According to the precedent shown with proline [105], we attempted to perform the reaction. Unfortunately, the aldol product did not exhibit optical activity. We cannot succeed in obtaining a material possessing a molecular chirality. This is a future challenge.

Controlling the helical direction in this approach remains a critical challenge because it is stochastic. Various scenarios for the emergence of homochirality have been explained by external and internal reasons existing on the primary Earth. Gravity is thought to be a strong candidate as the chiral inducer. A hypothesis about the similarity between Earth's movements, i.e., the Earth's right-handed rotation near the moon, alongside the right-handed Earth's orbital around the Sun, was reported [106–110]. Furthermore, the handedness of the chirality was reported to depend on the mechanical rotational direction [18]. Therefore, it seems to be possible to control the handedness by using such a macroscopic rotation. Very recently, the organic analysis of the samples corrected from the surface of asteroid Ryugu revealed the presence of both *L*- and *D*-amino acids [111]. This finding suggests a possibility that the primary Earth is not the homochiral world: amino acids with opposite handedness coexist in non-equal populations. This chiral conglomerate state in Ryugu is similar to the first step of our strategy. We propose the possibility that the dominant chirality is spread with a chiral amplification system to the world on Earth, and thus the emerged homochirality is transferred to the next generation.

We have demonstrated how chirality occurs in an achiral system and how the chirality is transferred to a higher-ordered architecture by using the liquid-crystalline system. These mechanisms not only provide a new approach for producing chiral materials without a chiral component but also a better understanding of the origin of homochirality in life.

**Supplementary Materials:** The following supporting information can be downloaded at: https://www.mdpi.com/article/10.3390/cryst14010097/s1, Video S1: Optical textures of the Iso—DC transition without and with the polymer network.

**Funding:** This research received no external funding.

**Data Availability Statement:** All of the data and methodologies associated with the research described are available through the references.

**Acknowledgments:** The author would like to thank his collaborators named in the reference list. The author would like to thank Royal Society of Chemistry for the reproduction of figures from the cited papers.

**Conflicts of Interest:** The author declares no conflicts of interest.

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
