# Peer review of "The Formation of Supramolecular Chiral Materials from Achiral Molecules Using a Liquid-Crystallin System: Symmetry Breaking, Amplification, and Transfer"

_crystals, doi:10.3390/cryst14010097_

Round 1
Reviewer 1 Report
Comments and Suggestions for Authors
In this manuscript, the author introduces the research progress on chirality in liquid crystal (LC) supramolecules in his group, taking as a starting point how chirality arises. Based on the design concept of LC supramolecules, the author was constructed the chiral structures in LC materials (including LC themselves, external field-induced LC materials, polymers) using the theory of spontaneous mirror symmetry breaking. Using trimers as an example, the author describes the formation, the transfer and the amplification of chirality in molecules by spatial structure (layer chirality and conformational chirality) in detail. This review has better implications for buliding the chiral structures. I agree to publish the manuscript in present form.
Author Response
I would like to thank the reviewer very much for the very favorable comments.
Reviewer 2 Report
Comments and Suggestions for Authors
This manuscript systematically summerized the different methods of introducing the supramolecular chirality in addtion to the symmtery breaking of liquid crystals, described the spontaneous symmetry breaking in the achiral liquid crystal compounds, external field induced mirror symmetry breaking in liuqid crystals, and the formation of helical polymers without a chiral component. Overall, it is well organized. It can be aceppted after addressing the following issues.
1. What is the meaning of "glassy nematic phase" and "grassy chiral nematic phase"?
2. What does the abbreviation DC stand for? The full name should be stated.
3. In the first line of Page 15, "The Figure 23" in the text should changed to "Figure 24".
4. For the samples II-n series, when the temperatures are below 100 ℃, what are the phase struture? and what is the treatment temperatures for the samples in the SEM images of Figure 24-26? In other words, whether the phase structure of the samples are in the DC phase at this time?
5. In Figure 21, the chemical structure of II-n is incomplete, please revise it.
6. There are some misspellings or syntax errors, please pay attention to the expressions.
Comments on the Quality of English LanguageIt can be improved further.
Author Response
I would like to thank the reviewer for the very favorable evaluation and the helpful comments. I have revised the manuscript according to the reviewer’s comments.
Reviewer’s comment
- What is the meaning of "glassy nematic phase" and "grassy chiral nematic phase"?
Author’s reply
I have added the following explanation for the glassy phases into the text.
“The glassy nematic and glassy chiral nematic phases are a glassy phase with a nematic-like orientational order and that with a chiral nematic-like orientational order, respectively. Both phases have a positional order.”
Reviewer’s comment
- What does the abbreviation DC stand for? The full name should be stated.
Author’s reply
I have stated the full name as follows.
“We call this dark chiral conglomerate phase as DC (dark chiral conglomerate) phase.”
Reviewer’s comment
- 3. In the first line of Page 15, "The Figure 23" in the text should changed to "Figure 24".
Author’s reply
I thank the reviewer very much for the helpful comment. I have found that numbers of many figures were incorrect. These are due to my careless mistakes. I have renumbered them correctly.
Reviewer’s comment
- For the samples II-n series, when the temperatures are below 100 ℃, what are the phase structure? and what is the treatment temperatures for the samples in the SEM images of Figure 24-26? In other words, whether the phase structure of the samples are in the DC phase at this time?
Author’s reply
The samples II-n series exhibited the DC phase at room temperature. I have added the following sentence.
“The super cooling DC phases of trimers II-n were stable at room temperature as same as those of trimers I-n.”
The treatment temperatures for the samples in the SEM images were room temperature. I have added the following sentence to each figure caption.
“The SEM measurements were carried out at room temperature.”
I think that the phase structure in the DC phase at room temperature is almost the same as that in the high temperature region. Unfortunately, we have never observed the temperature-dependent SEM images, so we could not confirm it.
Reviewer’s comment
- In Figure 21, the chemical structure of II-n is incomplete, please revise it.
Author’s reply
I think that this error occurred during the editing process. I will check the edited manuscript carefully.
Reviewer’s comment
- There are some misspellings or syntax errors, please pay attention to the expressions.
Author’s reply
I thank the reviewer again. I have corrected those errors and considered the expressions carefully.
Reviewer 3 Report
Comments and Suggestions for Authors
Ref.comments to the paper titled as “The Formation of Supramolecular Chiral Materials from an Achiral Molecules using a Liquid-Crystallin System: Symmetry Breaking, Amplification, and Transfer” written by Atsushi Yoshizawa.
Currently, more and more scientific and engineering groups are paying attention to the creation of the novel materials based on the liquid crystal (LC) mesophase. It is due to the fact, that the materials in this phase can reveal the anisotropic features activated by the electric, magnetic, laser, acoustic, thermal, etc. field. From this point of view the manuscript is actual and modern.
For the first, this review is good constructed and included the synthetic process, schematic models, analytical presentations, good explanation of the properties of the novel LC composites. Moreover, the papers published last 5 years have been considered as well. But, some interesting and important phenomena have not been analyzed in this review.
Firstly, I am recommending to author to consider and include in the paper the references from Prof.Chigrinov team: the photo aligning process used in the LC devices, LC for Photonics area, etc.
1). Chigrinov VG, Kozenkov VM, Kwok HS. Photoalignment of liquid crystalline materials: physics and applications. John Wiley & Sons; 2008 Sep 15.
2). Chigrinov VG, Kudreyko AA, Kozenkov VM. Kinetics of photoinduced phase retardation in azo dye layer. Liquid Crystals. 2022 Feb 9:1-8.
3). Vladimir G. Chigrinov, Liquid Crystal Photonics, 165 pp., Nova Science Publishers, December 2014. https://www.amazon.com/Liquid-Crystal-Photonics-Engineering-Techniques/dp/162948315X/ref=sr_1_6?s=books&ie=UTF8&qid=1488254729&sr=1-6&keywords=chigrinov
Secondly, Please include in your review the analysis of the papers of Prof.Galyametdinov team. So many LC structure can be modified via introduction of the quantum dots.
1). K. A. Romanova, Yu. G. Galyametdinov. Quantum-Chemical Simulation of Optical Functional Materials Based on Semiconducting Quantum Dots CdSe/CdS and Liquid-Crystalline Polymers. June 2020, DOI:10.18083/LCAppl.2020.2.76
2). Maksim Karyakin,… Yu. G. Galyametdinov. Liquid Crystalline Mixtures Consisting of Mesogenic Gadolinium Complex and Nematic Liquid Crystals, December 2019, DOI:10.18083/LCAppl.2019.4.67
3). A. A. Leshcheva, …, Yu. G. Galyametdinov. Thermosensitive material based on europium(iii) mesogenic complex. January 2022, Herald Of Technological University 25(11):32-35. DOI:10.55421/1998-7072_2022_25_11_32
Thirdly, Please include in your consideration the analysis of the papers from Prof.Kamanina team: about, for example, the formation of the 3D LC media under the condition of the laser interaction with the nanostructured LC; about the nanostructured relief influence on the orientation of the LC, etc.
1). N.V. Kamanina, “Fullerene-dispersed liquid crystal structure: dynamic characteristics and self-organization processes” Physics-Uspekhi 48 (4), 419-427 (2005). DOI:10.1070/PU2005v048n04ABEH002101
https://doi.org/10.1070/PU2005v048n04ABEH002101
2). Kamanina, N. Refractive Properties of Conjugated Organic Materials Doped with Fullerenes and Other Carbon-Based Nano-Objects. Polymers 2023, 15, 2819. 14 pages. https://doi.org/10.3390/polym15132819.
3). Kamanina, N.; Toikka, A.; Barnash, Y.; Zak, A.; Tenne, R. “Influence of Surface Relief on Orientation of Nematic Liquid Crystals: Polyimide Doped with WS2 Nanotubes” // Crystals 2022, 12(3), 391. 9 pages. https://doi.org/10.3390/cryst12030391. https://www.mdpi.com/journal/crystals.
As for my general local opinion: The paper is interesting and prepared with good illustrations and table data. It can be interesting in order to collect our knowledge in the LC media properties and application. Moreover, it can be useful for the education process as well.
Thus, the paper can be published after minor corrections.

Author Response
I would like to thank the reviewer for the favorable comments and the important suggestions.
The reviewer notes that some interesting and important phenomena have not been analyzed in this review. I agree with the reviewer on this comment. On the other hand, this review focuses on “the formation of supramolecular chirality in liquid crystals”. Unfortunately, I cannot include the other important phenomena not directedly related to chirality. From this point of view, I have revised the manuscript according to the reviewer’s suggestions.
Prof. Chigrinov’s team reported important researches on the photoalignment of liquid crystals with azo materials. Therefore, I have introduced them in the first paragraph of the section 4 as follows.
“Azobenzene derivatives are expected to be useful for applications in photomemory [72,73], photoalignment [74–79], or photomobile [80,81].”
References
- Chigrinov, V. G.; Kozenkov, V. M.; Kwok, H.-S. Photoalignment of Liquid Crystalline Materials: Physics and Applications, John Wiley & Sons, 2008, pp. 1–248.
- Chigrinov, V. G.; Kudreyko, A. A.; Kozenkov, V. M. Kinetics of photoinduced phase retardation in azo dye layer. Liquid Crystals 2022,49, 1376–1383.
- Chigrinov, V. G. Liquid crystal applications in photonics. SID 2016, 47(1), 927–936.
I am sorry to say that I cannot introduce the interesting and important researches from Prof.Galyametdinov’s team and Prof.Kamanina’s team because they are not within the scope of this review.
Reviewer 4 Report
Comments and Suggestions for Authors
The presented review is devoted to the issues of chirality of low-molecular compounds and polymer structures. The review covers 106 literary sources and contains a large number of illustrations that help to better understand the presented material. For illustrations, I would suggest that authors increase the font size. In some places the inscriptions are poorly perceived. In the manuscript, the authors provide a number of examples demonstrating the influence of external influences on the formed chiral structure, which is important to take into account during the synthesis process.
The review contains a large number of errors and typos that need to be corrected before publication.
I recommend that the author change the Abstract by excluding from it the listing of the main parts of the review and concentrate on the results achieved.
Line 103. Typo - "ptical"
Line 115. "solovents" - perhaps here I will enter "solvents"
Line 135. "asymmtric" is a typo
Line 138. "super molecules"
Line 140. "supramolecule"
Line 146. perhaps "water" is redundant here.
Line 155. "electricl"
Line 157. "Moleucules"
Line 174. "short-rane"
Line 200. "supramorcular"
Line 616. "filed"
Line 774. "regaeded"
Figure 40. The image for the amorphous phase - BPIII is a little confusing. The depicted structure has a general preferred direction, but for the amorphous phase would a lower order be observed?!
Figure 51. The resulting film has a slightly strange shape?!
You need to check the links, in particular link 80.
Overall, the review is of interest and may be considered further by the editor.
Comments on the Quality of English LanguageThe presented review is devoted to the issues of chirality of low-molecular compounds and polymer structures. The review covers 106 literary sources and contains a large number of illustrations that help to better understand the presented material. For illustrations, I would suggest that authors increase the font size. In some places the inscriptions are poorly perceived. In the manuscript, the authors provide a number of examples demonstrating the influence of external influences on the formed chiral structure, which is important to take into account during the synthesis process.
The review contains a large number of errors and typos that need to be corrected before publication.
I recommend that the author change the Abstract by excluding from it the listing of the main parts of the review and concentrate on the results achieved.
Line 103. Typo - "ptical"
Line 115. "solovents" - perhaps here I will enter "solvents"
Line 135. "asymmtric" is a typo
Line 138. "super molecules"
Line 140. "supramolecule"
Line 146. perhaps "water" is redundant here.
Line 155. "electricl"
Line 157. "Moleucules"
Line 174. "short-rane"
Line 200. "supramorcular"
Line 616. "filed"
Line 774. "regaeded"
Figure 40. The image for the amorphous phase - BPIII is a little confusing. The depicted structure has a general preferred direction, but for the amorphous phase would a lower order be observed?!
Figure 51. The resulting film has a slightly strange shape?!
You need to check the links, in particular link 80.
Overall, the review is of interest and may be considered further by the editor.
Author Response
I would like to thank the reviewer for the favorable comments and the helpful suggestions. I have revised the manuscript according to the reviewer’s comments.
Reviewer’s comment
For illustrations, I would suggest that authors increase the font size. In some places the inscriptions are poorly perceived.
Author’s reply
I have increased the font size in Figure 7 of the revised manuscript. Some figures have been reseized
Reviewer’s comments
The review contains a large number of errors and typos that need to be corrected before publication. You need to check the links, in particular link 80.
Author’s reply
I would like to thank the reviewer for the helpful comments very much. I have found a lot of errors in both the text and the references. I have corrected them carefully.
Reviewer’s comment
I recommend that the author change the Abstract by excluding from it the listing of the main parts of the review and concentrate on the results achieved.
Author’s reply
According to the reviewer’s comment, I have revised the Abstract as follows.
“Recently, the formation of chiral architectures by self-organization of achiral small molecules has been intensively investigated. How does the chirality arise? How is the chirality transferred to a higher ordered structure? We will describe recent research developments in supramolecular chirality in liquid crystals focusing primarily on our group’s experimental results. We show the following concepts in this review. Spontaneous mirror symmetry breaking in self-assembled achiral trimers under a nonequilibrium state induces the supramolecular chirality. Two kinds of domains with opposite handedness exist in non-equal population. If the dominant domain can be amplified to produce a homochiral template, the chirality is transferred to a polymer film in the course of polymerization of achiral reactive monomers by using the template. Finally, we discuss how the concepts obtained from this liquid crystal research relate to the origin of homochirality in life.”
Reviewer’s comment
Figure 40. The image for the amorphous phase - BPIII is a little confusing. The depicted structure has a general preferred direction, but for the amorphous phase would a lower order be observed?!
I thank the reviewer for the important comment. I have replaced the image of the BPIII in Figure 41 of the revised manuscript with that possessing a lower order.
Reviewer’s comment
Figure 51. The resulting film has a slightly strange shape?!
Author’s reply
This picture is a photograph of the film on a glass slide. I think that it looks a slightly strange shape. I have revised both the text and the figure caption as follows.
“Figure 52 shows the schematic illustration of the preparation process of the photo-polymerized film, a photograph of the polymer film on a glass plate, and the FE-SEM images of the surface structures of the PS-DC material and the polymer film.